# Selectivity to approaching motion in retinal inputs to the dorsal visual pathway

**Todd R Appleby[1,2,3], Michael B Manookin[2,3]***

[1]Graduate Program in Neuroscience, University of Washington, Seattle, United States; [2]Department of Ophthalmology, University of Washington, Seattle, United States; [3]Vision Science Center, University of Washington, Seattle, United States

**Abstract** To efficiently navigate through the environment and avoid potential threats, an animal must quickly detect the motion of approaching objects. Current models of primate vision place the origins of this complex computation in the visual cortex. Here, we report that detection of approaching motion begins in the retina. Several ganglion cell types, the retinal output neurons, show selectivity to approaching motion. Synaptic current recordings from these cells further reveal that this preference for approaching motion arises in the interplay between presynaptic excitatory and inhibitory circuit elements. These findings demonstrate how excitatory and inhibitory circuits interact to mediate an ethologically relevant neural function. Moreover, the elementary computations that detect approaching motion begin early in the visual stream of primates.

## Introduction

As an object approaches, the image of that object becomes larger on the surface of an observer's retina. Many animals use these size changes to estimate whether and when an object will collide with the animal (*Schiff and Detwiler, 1979*; *Lee, 1976*; *Kaiser and Hecht, 1995*) and also to estimate the animal's own motion through the environment (*Clifford et al., 1999*; *Schrater et al., 2001*). Neurons with such selectivity for approaching motion have been found in the dorsal visual pathway of primates (*Orban et al., 1992*; *Duffy and Wurtz, 1991*; *Wang and Yao, 2011*). However, it is not known whether similar approach selectivity is found earlier in the visual pathway of primates.

Several ganglion cell types found in the retinas of humans and non-human primates project to the dorsal visual pathway, including parasol (magnocellular-projecting) cells (*Rodieck and Watanabe, 1993*). These cells can detect small changes in the reflectance of an object relative to the background (i.e. contrast), and their high contrast sensitivity has resulted in the hypothesis that these cells contribute primarily to representations of object form (*Kaplan and Shapley, 1986*; *Lee et al., 1995*). Less is known about how motion affects the response properties of these cells (*Chichilnisky and Kalmar, 2003*; *Frechette et al., 2005*; *Manookin et al., 2018*).

Here, we report that parasol and other ganglion cell types in the macaque monkey retina display a preference for approaching motion. We show that visual circuits downstream of parasol cells can detect approaching motion based solely on the spike output of these cells. We further study the synaptic basis for this computation using direct recordings and a computational model. In summary, the elementary computations for detecting approach are present in the retinal input to the dorsal visual pathway of primates.

## Results

We recorded the spike responses of five ganglion cell types—broad thorny, On and Off smooth monostratified, and On and Off parasol ganglion cells—in an in vitro preparation of the macaque monkey retina to determine whether these cells showed preference for approaching or receding

*For correspondence:
manookin@uw.edu

Competing interests: The authors declare that no competing interests exist.

motion. Cells were identified based on their characteristic cell body sizes and shapes under infrared illumination, their distinct light response properties, and dendritic morphologies (*Puller et al., 2015*; *Watanabe and Rodieck, 1989*; *Crook et al., 2008*; *Rhoades et al., 2019*; *Petrusca et al., 2007*). We begin by demonstrating that these cells show strong selectivity for approaching textures.

## Five primate ganglion cell types show preference for approaching textures

Humans and non-human primates use changes in spatial scale or object size in estimating the speed of approaching objects and their own motion through the environment (*Schiff and Detwiler, 1979*; *Lee, 1976*; *Kaiser and Hecht, 1995*; *Clifford et al., 1999*; *Schrater et al., 2001*). Further, this selectivity for approaching motion is commonly thought to originate in the thalamus or visual cortex of primates (*Wang and Yao, 2011*), and a retinal origin for this type of motion sensitivity has not been considered.

To determine whether ganglion cells in the primate retina exhibited a preference for approaching motion, we recorded cellular responses to moving stochastic textures (see Materials and methods). The spatial scale of the textures changed as a function of time—increasing in scale to simulate approaching motion or decreasing in scale to simulate receding motion. Further, both stimulus classes contained the same ranges of images scales—the receding motion stimuli were simply the time-reversed image sequences of the approaching stimuli (*Figure 1B*; *Schrater et al., 2001*; *Wang and Yao, 2011*). Further, we tested the same range of scale changes that have been shown to elicit percepts of approaching motion in humans (*Schrater et al., 2001*).

As shown in *Figure 1*, approaching textures elicited larger spike responses than receding textures in parasol, smooth monostratified, and broad thorny (On-Off type) ganglion cells. A large bistratified ganglion cell–another type of On-Off cell—also showed increased spiking to approaching textures, but the bias to approaching motion was not as strong as that observed in the other cell types (*Figure 1A*, *bottom*). This finding indicated that preference for approaching motion was not a universal property of all primate ganglion cells, but was restricted to a subset of these cell types.

We quantified the degree to which a cell preferred approaching or receding motion by calculating the difference between the spike responses to approaching ($R_{approaching}$) and receding ($R_{receding}$) motion divided by the sum of those responses, as described by *Equation 1*.

$$approach\ selectivity = \frac{R_{approaching} - R_{receding}}{R_{approaching} + R_{receding}} \tag{1}$$

Positive values indicate a preference for approaching motion and negative values for receding motion while values near zero indicate a lack of preference. Indeed, the approaching textures elicited higher spike rates in On- and Off-type parasol and smooth monostratified ganglion cells, resulting in significant approach selectivity values in all four cell types (*Figure 1C*; $p < 7.8 \times 10^{-3}$; Wilcoxon signed rank test).

The smallest texture scales used in these experiments corresponded in size to the dendritic tree diameters of the diffuse bipolar cells that provide excitatory synaptic input to parasol and smooth monostratified ganglion cells in the mid-peripheral retina ($\sim$30–40 µm) (*Dacey et al., 2000*; *Boycott and Wässle, 1991*; *Tsukamoto and Omi, 2015*; *Tsukamoto and Omi, 2016*; *Turner and Rieke, 2016*). However, to reliably contribute to vision, these mechanisms must operate across the broad range of spatial scales encountered in the natural environment (*Field, 1987*; *Dong and Atick, 1995*). To ensure that approach selectivity was not restricted to a limited subset of texture scales (i.e. spatial frequencies), we repeated the texture experiments using four distinct scale ranges in the same cell. The smallest scales in the texture sequences ranged from the approximate width of diffuse bipolar cells (3.3 cycles degree$^{-1}$) to the approximate dendritic tree width of parasol ganglion cells in the mid-peripheral macaque retina (0.8 cycles degree$^{-1}$).

Varying the scale ranges of the texture sequences did not change the preference for approaching textures. Approach selectivity persisted in parasol and smooth monostratified cells across these stimulus conditions (parasol: n = 16 cells, $p < 2.4 \times 10^{-4}$; smooth monostratified: n = 7 cells, $p < 7.8 \times 10^{-3}$; Wilcoxon signed rank test). In fact, approach selectivity increased with increasing texture scale (i.e. decreasing spatial frequency) in both parasol and smooth monostratified cells (*Figure 1D*). These results indicated that the mechanisms mediating approach selectivity in these cells operate across a

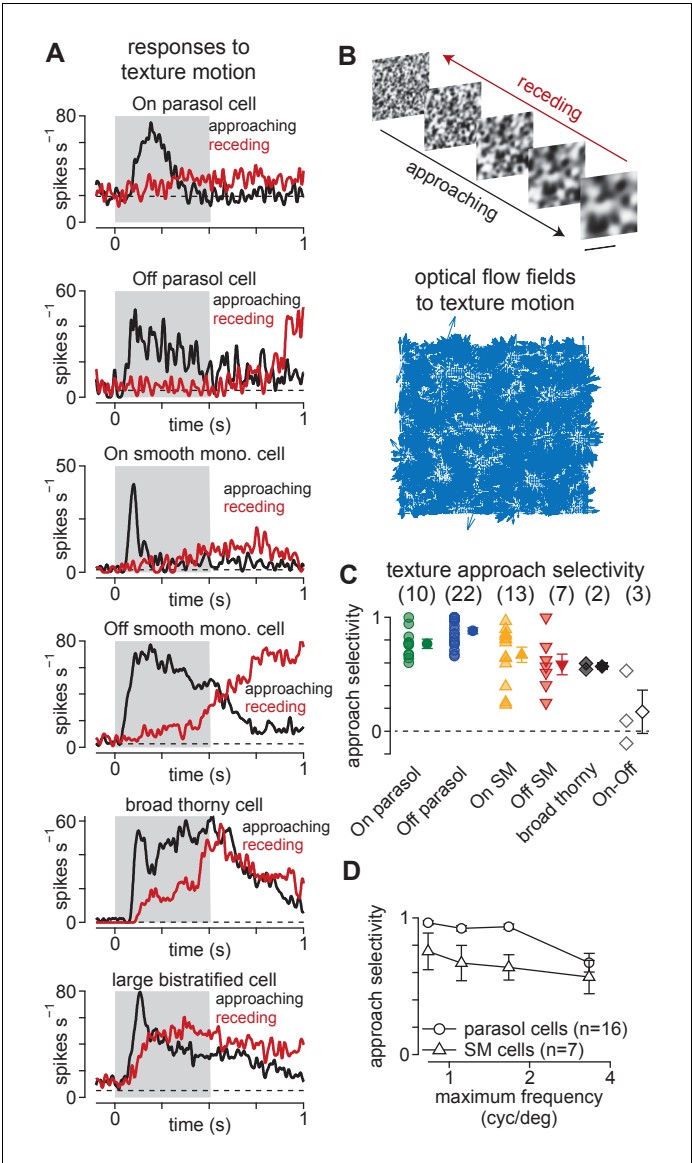

**Figure 1.** Ganglion cells exhibit a preference for approaching textures. (**A**) Responses of several ganglion cell types to receding (*red*) and approaching (*black*) Gaussian textures. Average spike rate is shown across 50–200 distinct randomly generated textures. The gray region indicates the period of motion. (**B**) Example of receding and approaching texture stimuli used in the experiments (*top*). Scale bar indicates 0.5 mm. *Bottom*, Optical flow fields computed from an example approaching texture movie. White areas show regions from which the texture expanded during the stimulus sequence. (**C**) Approach selectivity index values for cell types in (**A**) to the texture stimuli. Transparent shapes indicate individual cells. Opaque shapes and error bars indicate mean ± SEM. (**D**) Approach selectivity (*y*-axis) as a function of the initial spatial frequency (*x*-axis) in parasol (n = 16) and smooth monostratified ganglion cells (n = 7). Approach selectivity persisted at all spatial frequencies tested.
The online version of this article includes the following source data for figure 1:

**Source data 1.** Included is a data file containing a structure for the approach selectivity data in *Figure 1*.

wide range of spatial scales. Further, the texture stimuli contained similar statistical properties to the types of motion encountered during optical flow, indicating that signals from parasol and smooth monostratified cells could be utilized in detecting this type of motion.

## Approach motion selectivity predicted from nonlinear subunits

Neurons in macaque visual cortex show a preference for approaching textures, which is thought to arise from the linear spatiotemporal receptive-field structure of these cells (*Wang and Yao, 2011*). However, neural mechanisms operating much earlier in the visual pathway might also contribute to the observed preference for approaching motion.

We considered the excitatory synaptic output from retinal bipolar cells onto the dendrites of ganglion cells as a strong candidate for contributing to this type of stimulus selectivity. The synaptic output of retinal bipolar cells is strongly rectified (nonlinear) (*Demb et al., 1999*; *Schwartz et al., 2012*) and this rectification is critical for detecting certain types of visual motion (*Demb et al., 2001*; *Baccus et al., 2008*; *Kuo et al., 2016*; *Manookin et al., 2018*). Thus, we used a combination of computational modeling and synaptic current recordings to determine whether bipolar cells contributed to approach selectivity in parasol and smooth monostratified cells.

We created subunit receptive-field models based on direct measurements of the spatiotemporal filtering properties and output nonlinearities of the diffuse bipolar cells that provide excitatory synaptic input to parasol and smooth monostratified ganglion cells in the macaque monkey retina. The spatial components of the model bipolar cell receptive fields were based on previous measurements of these cells (*Dacey et al., 2000*; *Boycott and Wässle, 1991*; *Tsukamoto and Omi, 2015*; *Turner and Rieke, 2016*) and the temporal filtering properties, electrical coupling, and output nonlinearities were based on our own direct measurements (*Manookin et al., 2018*).

The receptive-field profile of ganglion cells was modeled as a difference-of-Gaussians in which the receptive-field center was opposed by a surround of opposite contrast polarity. This receptive-field profile set the weighting of subunit inputs to model ganglion cells. The sizes and strengths of the center and surround regions of the receptive field were determined directly by measuring spike responses to sinusoidally modulated spots that varied in diameter (14–720 µm)—the relative sizes and strengths of center ($w_C$) and surround ($w_S$) regions of the receptive field were estimated from these response patterns (*Figure 2B, C*; see Materials and methods).

Both center and surround regions were comprised of subunits and the surround provided lateral inhibition to the model ganglion cell with a temporal delay (see Materials and methods). This delay occurs because surround inhibition typically arises via feedback from horizontal cells or amacrine cells and, thus, must traverse an extra synapse relative to the direct excitatory synaptic input from bipolar cells. We directly measured the surround delay by recording responses to spots presented over the receptive-field center or annuli presented in the surround. Spot or annulus contrast was drawn from a Gaussian distribution on each frame and the temporal filtering properties of center and surround regions were measured directly by cross-correlating the contrast trajectory of the stimulus with the cell's spike output (see Materials and methods). The temporal delay between center and surround regions was determined from the difference in the time-to-peak of the temporal filters measured in the center and surround regions of the receptive field (*Figure 2D, E*). This delay was then incorporated into the model at the level of the surround subunits.

To determine whether the subunit models predicted the observed approach selectivity, we obtained the model outputs to moving textures. To simulate approaching or receding motion, the spatial frequency content (i.e. spatial scale) of the textures changed as a function of time (*Schrater et al., 2001*; *Wang and Yao, 2011*). We compared two models that were identical except for the input-output function of the subunits—in one case the function was linear and in the other it was nonlinear. The outputs of the linear subunit model were similar for both approaching and receding textures, resulting in a lack of approach selectivity ($p > 0.4$ at all expansion rates; *Figure 3A*). However, the nonlinear subunit model showed a very different pattern—approaching textures produced significantly larger outputs than receding textures for each expansion rate and this bias for approaching motion increased with increasing rate (*Figure 3B*). These modeling results suggest that the nonlinear output of retinal bipolar cells may contribute to approach selectivity in some ganglion cell types (see Appendix 1 for a more detailed treatment of this subject).

These results are particularly important in the context of canonical models of motion sensitivity in primates. Previous studies posited that this type of approach selectivity arose in the cortex or thalamus (*Wang and Yao, 2011*). Our physiological recordings, however, indicated that some primate ganglion cell types showed approach selectivity. Moreover, our modeling results indicated that approach sensitivity was predicted as a consequence of ganglion cells pooling across several

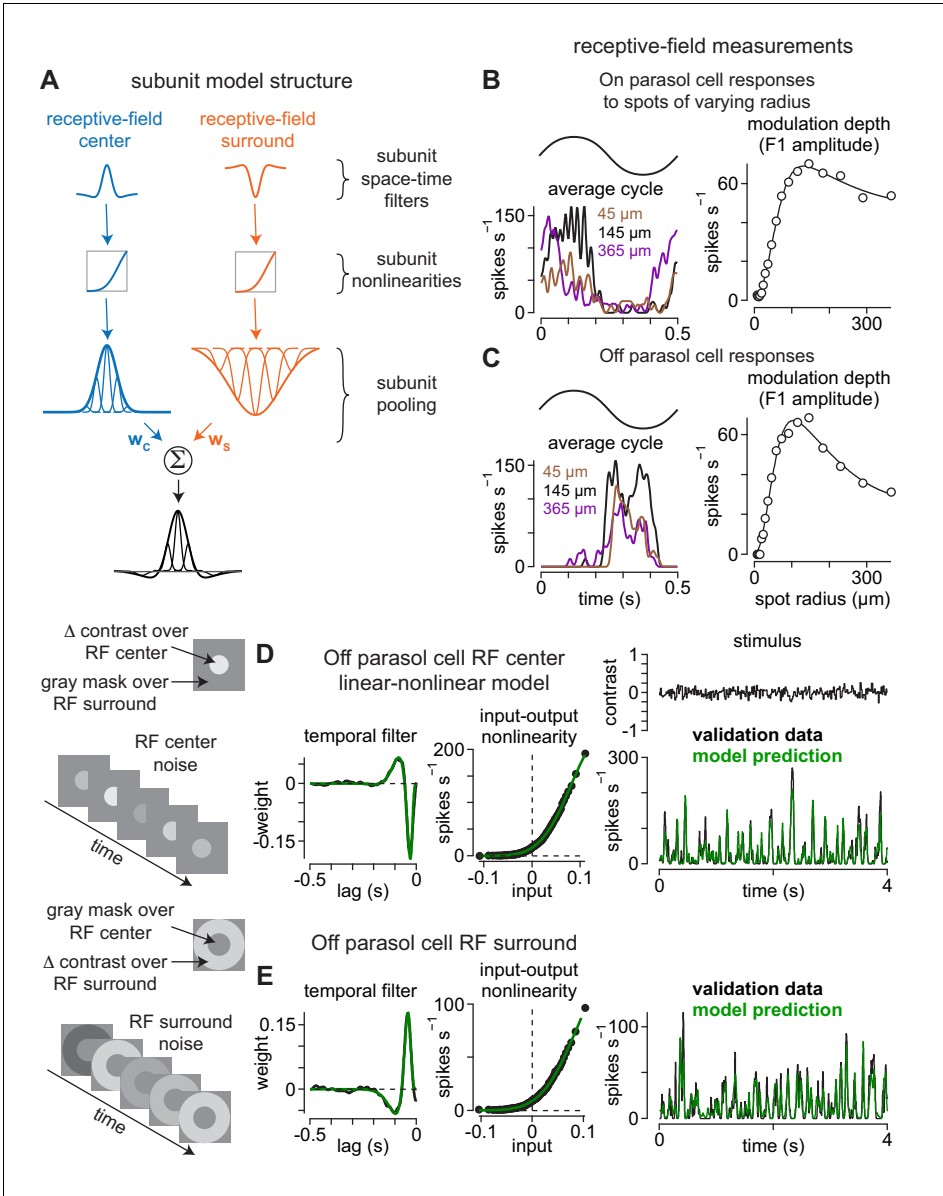

**Figure 2.** Direct measurement of spatiotemporal receptive-field model parameters. (**A**) Model architecture. Center (*left*) and surround (*right*) regions of the receptive field were comprised of subunits. Subunits also exhibited center-surround receptive field structures based on published measurements. Following spatiotemporal filtering, signals were passed through a static input-output nonlinearity after which they were normalized and integrated at the level of model ganglion cells. (**B**) The sizes and weightings of center ($w_C$) and surround ($w_S$) inputs to the model ganglion cells in (**A**) were calculated by recording spike responses to spots presented over an On parasol ganglion cell's receptive field. Spot contrast (0.5) was modulated sinusoidally at 2 Hz and spot radius varied on each trial. The modulation depth (F1 amplitude) of the average cycle was largest at 145 µm (*left*) and fell off at smaller and larger radii (*right*). Solid line shows difference-of-Gaussians fit to the data (*right*). (**C**) Same as (**B**) for an Off parasol cell. (**D–E**) The temporal lag between center (**D**) and surround (**E**) regions of the receptive field was measured using a Gaussian flicker stimulus. On each frame, the contrast of either a spot (center condition) or annulus (surround condition) was drawn randomly from a Gaussian distribution with a mean of 0.0 and a standard deviation of 0.1. Temporal filters were determined by cross-correlating the cell's spike output with the stimulus sequence (*left*) and the temporal lag between center and surround was determined from the time-to-peak of these filters. *Middle*, Input-output nonlinearities were determined for the center and surround noise. *Right*, Unique contrast sequences were interleaved with repeated sequences. The repeated sequences were not used in computing the temporal filters, but were used to cross-validate the model. The average response to the repeated sequences (*black*) showed high correspondence to the model prediction (*green*).

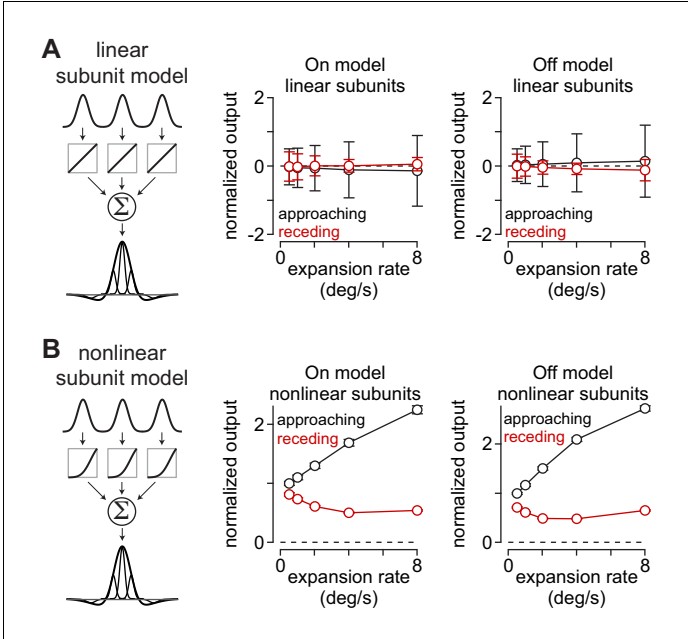

**Figure 3.** Approach selectivity to moving textures predicted from nonlinear receptive-field subunits. (**A**) Normalized outputs of linear subunit models to 500 textures that either approached or receded at five different rates. Approaching and receding motion was not distinguishable at any rate for either the On or the Off subunit models. (**B**) Outputs of models containing nonlinear subunits. Adding a nonlinearity at the model bipolar cell output produced selectivity for approaching textures at all expansion rates. Error bars indicate mean ± SEM.

nonlinear subunits. We will directly test the contribution of bipolar cells to approach motion selectivity later in this work, but first we examine whether retinorecipient brain regions could detect approaching motion from the outputs of parasol ganglion cells.

## Circuit model predicts selectivity for approaching motion

Our physiological recordings indicated that individual parasol cells could distinguish between approaching or receding texture motion with a high degree of accuracy (*Figure 1*). We employed our computational models to gain insight into how accurately the direction of moving textures could be inferred by downstream neurons from the outputs of populations of On and Off parasol cells.

Each subclass of parasol ganglion cell (On or Off) forms a regularly spaced mosaic with neighboring cells of the same class, but the dendritic-field and receptive-field locations between On and Off types are uncorrelated (*Watanabe and Rodieck, 1989*; *Field et al., 2010*). Thus, we randomly shifted the locations of the model On and Off cell mosaics relative to each other. We tested the model on 500 different textures moving at five different speeds (0.5–8 degrees s$^{-1}$). As with our direct recordings, the On and Off models showed larger responses for approaching textures relative to their receding counterparts (*Figure 4B*).

We tested linear and quadratic decoding models to estimate how accurately downstream neurons could distinguish between approaching and receding texture motion based on the outputs of the model On and Off parasol cells. The output of the linear decoding model was the scaled sum of the model parasol cell responses, and the quadratic model squared the outputs of these cells prior to scaling and summation (see Materials and methods). We assessed the models' ability to distinguish between approaching and receding motion by calculating the Jensen-Shannon distance between the model outputs to these stimuli (*Endres and Schindelin, 2003*; *Österreicher and Vajda, 2003*). This metric quantifies the degree of dissimilarity between the response distributions—values near zero indicate a high degree of similarity while values near one occur when the distributions are more distinct (see Materials and methods; *Equation 17*). Indeed, both decoding models showed Jensen-Shannon distance values ≥0.28 under all conditions and values near one at the higher speeds

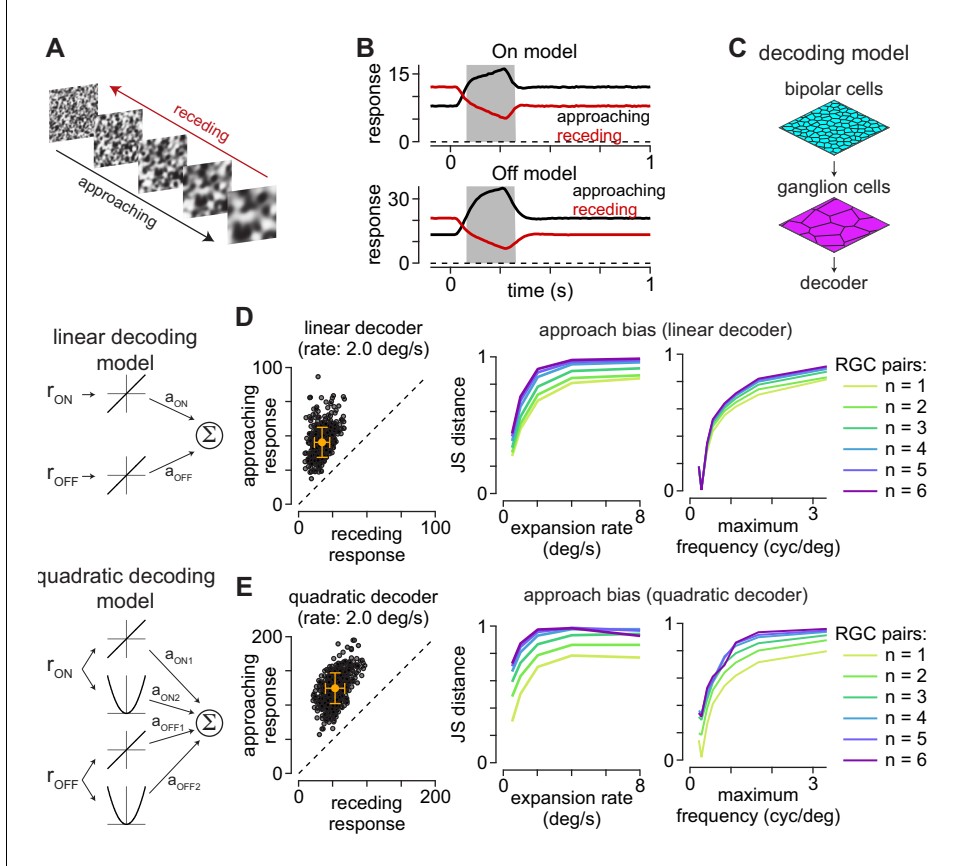

**Figure 4.** Computational model predicts selectivity for approaching textures. (**A**) Example approaching and receding textures. The same texture sequence was presented from highest-to-lowest spatial frequency (approaching) or lowest-to-highest spatial frequency (receding). Models were run on 500 unique texture sequences at five different expansion rates. (**B**) Average responses of On (*top*) and Off models (*bottom*) to 500 textures that approached or receded at 2.0 degree $s^{-1}$. (**C**) Basic organization of decoding models. Model bipolar cells provided input to ganglion cell mosaic which, in turn, provided input to the decoder. (**D**) The output of the linear decoding model was the weighted sum of the outputs from the On and Off models. The model produced larger outputs for approaching than receding motion for each of the textures, shown as individual circles (*left*). Jensen-Shannon distance values computed between the distributions for approaching and receding textures at five expansion rates (*middle*). This approach bias persisted across a range of expansion rates and was highest at higher rates. Lines are color coded for the number of ganglion cells combined by the decoder. *Right*, Jensen-Shannon distance (*y*-axis as function of the maximum spatial frequency in the moving textures (*x*-axis). Discriminability of the approaching textures persisted across a broad range of spatial frequencies and fell off when spatial frequency content was very low (<0.3 cycles degree$^{-1}$). (**E**) Same as (**D**) for the quadratic decoding model.

tested (*Figure 4D, E*). These results indicated that downstream circuits could accurately detect approaching motion with very simple processing of the inputs from On and Off parasol cells.

To determine whether decoding accuracy varied with the number of ganglion cell outputs, we calculated model performance while varying the number of On/Off output pairs (1–6 pairs). Pairings were established by calculating the nearest Euclidean distance between neighboring cells. Increasing the number of model ganglion cell pairs providing input to the linear decoder produced a modest increase in performance (*Figure 4D*). The effect on the quadratic decoder was more varied—increasing the number of ganglion cells greatly improved performance at the slowest expansion rates, but the effects were inconsistent at faster expansion rates. These results are consistent with the premise that integrating over a larger number of retinal outputs improved performance in detecting approaching textures. However, model performance was high for single ganglion cell pairs, indicating that little integration or post-processing of the retinal output was required to reliably detect approaching motion.

The type of motion that we studied with the approaching and receding texture stimuli is commonly encountered as an animal moves through the environment (*Schrater et al., 2001*; *Dong and Atick, 1995*). The results presented thus far in our study support the premise that the output of the primate retina contains reliable information that could be used by downstream neural pathways for detecting approaching motion. Our next goal was to understand the specific neural circuit mechanisms mediating the observed motion preference.

## Approach motion selectivity present for moving annuli

The subunit model in *Figure 3* demonstrated that the nonlinear input-output properties of bipolar cell subunits could account for much of the observed approach selectivity to texture motion. We next wanted to determine whether this output nonlinearity accounted for all the observed effect or whether other circuit mechanisms also contributed. We did this by designing a stimulus paradigm which should not elicit approach selectivity for the nonlinear subunit circuit motif described in *Figure 3*. Thus, if approach selectivity were observed, other circuit mechanisms must also contribute. The stimulus used was a ring (annulus) that expanded outwardly (approaching) or contracted inwardly (receding) along the dendritic tree (*Figure 5A*). *Figure 5B* illustrates the widest extent of the annulus relative to the receptive-field profiles of example On and Off parasol cells that were measured using the stimulus paradigm described in *Figure 2*. The annulus was contained within the receptive-field center, which allowed us to probe cellular and model response properties without strongly engaging the surround.

We tested the predictions of three model configurations on the approaching and receding annuli: (1) a model in which the subunit outputs were linear (linear subunit model), (2) a model in which the output of each subunit was passed through an output nonlinearity (nonlinear subunit model), and (3) a model in which simulated electrical coupling between subunits occurred prior to the output nonlinearity (coupled subunit model).

Unlike their predictions for the texture stimuli (*Figure 3*), the linear and nonlinear subunit models both predicted a lack of approach selectivity for the moving annuli, as model outputs were similar for the approaching and receding annuli at all contrasts (*Figure 5C, D*). The coupled subunit model, however, predicted larger responses to approaching annuli than to receding annuli (*Figure 5E*).

What accounts for the approach bias predicted by the coupled subunit model? It was proposed to us that the effect of coupling in expanding the subunit receptive-field size could explain the observed approach motion bias. Indeed, some studies have proposed that electrical coupling significantly increases the size of bipolar cell receptive fields (*Dacey et al., 2000*; *Kujiraoka and Saito, 1986*; *Saito and Kujiraoka, 1988*), but see *Berntson and Taylor (2000)*. We sought insight into whether altering the subunit receptive-field size could account for the observed approach bias. We did this by varying the subunit receptive-field diameters in models lacking electrical coupling between subunits.

We used a receptive-field diameter of 32 µm in our coupled subunit model that exhibited approach selectivity to moving annuli (two-standard-deviation diameter; *Figure 5E*). This value was based on a previous study in which we used direct measurements of excitatory synaptic currents in parasol ganglion cells to determine the subunit size, coupling gain between subunits (gain, 0.1), and space constant for electrical coupling in the diffuse bipolar cell networks (λ, 36.4 µm) (*Manookin et al., 2018*). These model parameters would expand the subunit receptive field by ~7 µm. Thus, if approach selectivity were a result of subunit receptive-field expansion, we would expect to observe comparable approach selectivity values for the coupled model and a model lacking electrical coupling with subunit diameters of ~39–40 µm. However, this was not the case (*Figure 5G, H*). The coupled subunit models showed approach selectivity values of ~0.5 when the coupling gain was 0.1 (On model, 0.56; Off model, 0.46), but the same models that lacked coupling and with subunit diameters of 40 m exhibited much lower approach selectivity (enlarged subunit model: On model, 0.09; Off model, 0.03). In fact, doubling the subunit receptive-field diameters did not reproduce the level of approach selectivity observed in the coupled subunit model or in our direct recordings from parasol ganglion cells (*Figure 5G, H*). Thus, approach bias does not arise primarily from enlarging subunit receptive fields.

These modeling simulations produced two principal results. First, the nonlinear subunit model predicted approach selectivity to the expanding texture stimuli (*Figure 3*), but not to the expanding

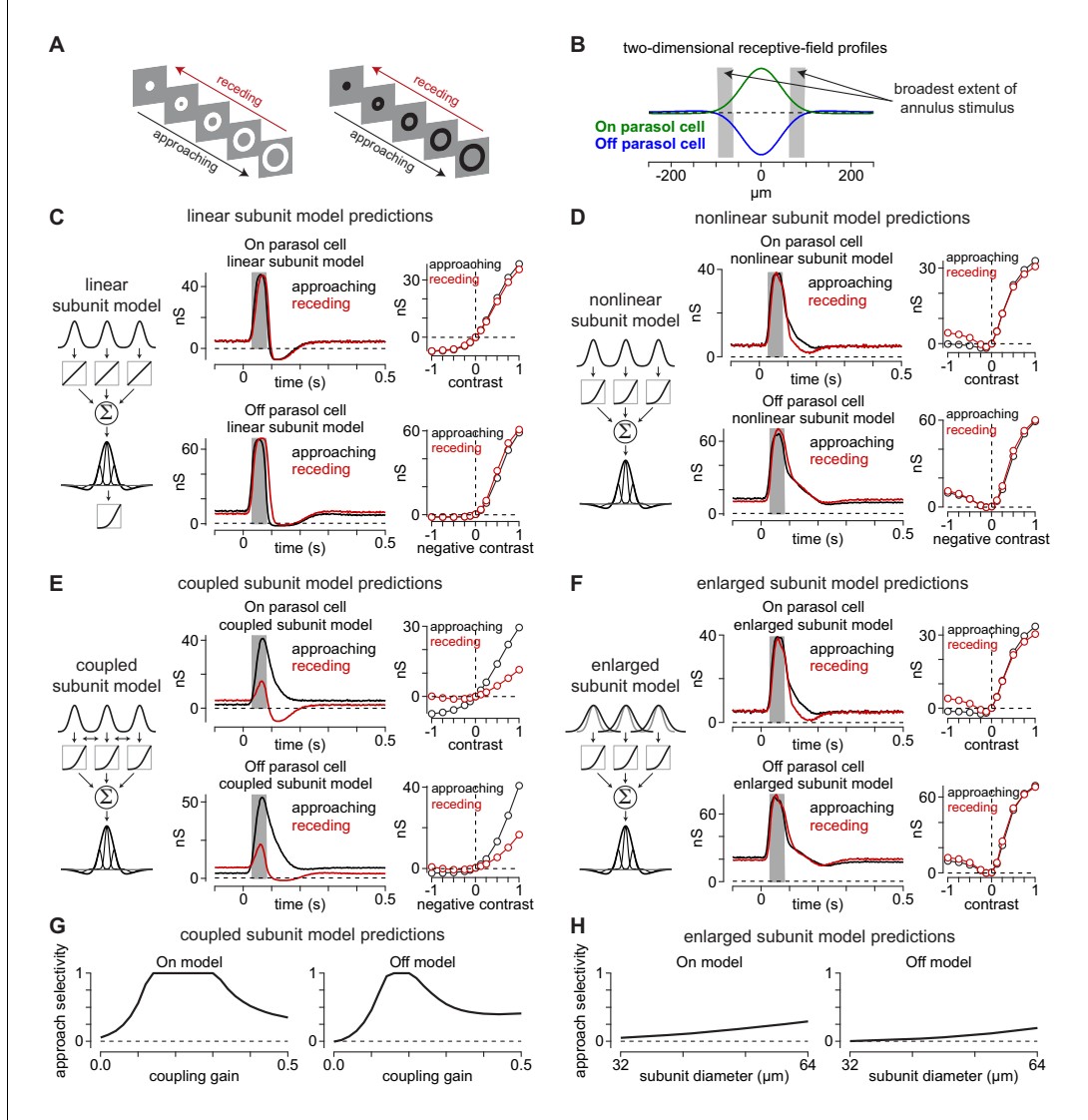

**Figure 5.** Canonical receptive-field models predict a lack of approach selectivity. (A) Stimulus paradigm for approaching and receding annuli. Annuli either rapidly increased in radius (*approaching*) or decreased in radius (*receding*). (B) Two-dimensional spatial receptive-field profiles used in the model. Values were determined directly from parasol cell recordings as shown in *Figure 2*. Gray regions indicate the widest extent of the moving annuli. (C) The linear subunit model was created in which subunit input was integrated linearly prior to a single nonlinearity at the ganglion cell output. Model parameters came directly from measurements of spatiotemporal receptive field properties. On and Off models predicted similar responses for both approaching and receding annuli. (D) A model in which the output of each subunit was passed through the output nonlinearity prior to integration at the level of the ganglion cell also predicted similar responses to approaching and receding annuli. The nonlinear subunit model also predicted similar responses in Off parasol cells to approaching and receding annuli of a given contrast. (E) Output of On and Off models with electrical coupling between bipolar cells. This models produced excitatory conductances that were biased toward approaching motion. (F) A model in which coupling between subunits was absent, but the subunit receptive-field sizes were enlarged to simulate lateral spread through gap junctions. Approach selectivity was absent from this model, indicating that the increase in subunit receptive-field size alone could not account for the approach selectivity observed in (E). (G) Approach selectivity index calculated for the On and Off cell coupled subunit models at a series of coupling gains. Approach selectivity was highest for gains of ~0.1–0.4. (H) Approach selectivity for the enlarged subunit models calculated for a series of subunit diameters. Selectivity was relatively low even at large diameters.

The online version of this article includes the following video, source code and figure supplement(s) for figure 5:

**Source code 1.** Included is a MATLAB file containing code used to generate a simplified coupling model.

**Figure supplement 1.** Parasol cell responses to briefly flashed annuli.

**Figure 5—video 1.** Movie illustrating model subunit activations during the annulus stimulus paradigm.

https://elifesciences.org/articles/51144#fig5video1

annuli. Thus, if this model accurately reflects the underlying circuitry, then we should observe a lack of approach selectivity to the annulus stimulus in our direct recordings from primate ganglion cells.

Another important result of the simulations was the divergent predictions of the nonlinear and coupled subunit models (*Figure 5D, E*). These models were identical other than that the latter model simulated electrical coupling between subunits prior to the output nonlinearity. Thus, the differences in predicted output pattern between these two models suggested that electrical coupling in the bipolar cell network contributes to approach bias for certain classes of stimuli such as the moving annuli (see Appendix 1). If such a mechanism were engaged by these stimuli, we would expect to observe approach selectivity to the moving annuli in our cellular recordings. Indeed, we next tested these predictions by recording the responses of ganglion cells to the moving annulus stimulus paradigm.

Consistent with the predictions of the coupled subunit model, parasol cells exhibited pronounced asymmetries to approaching and receding annuli. In On parasol cells, approaching bright annuli evoked much larger spike responses than receding annuli of the same contrast—at the highest contrast, approaching motion elicited $85.4 \pm 10.9$ spikes $s^{-1}$ versus $7.1 \pm 4.3$ spikes $s^{-1}$ for receding motion (contrast, +1.0; mean $\pm$ SEM; n = 26 cells; p=$8.3 \times 10^{-6}$; p<$2.6 \times 10^{-5}$ at all contrasts; Wilcoxon signed rank test, here and below). Likewise, approaching motion to strong negative contrasts in Off parasol cells evoked $148.7 \pm 12.8$ spikes $s^{-1}$ versus $55.2 \pm 9.4$ spikes $s^{-1}$ for receding motion (contrast, –1.0; mean $\pm$ SEM; n = 24 cells; p=$2.1 \times 10^{-5}$; p<$3.4 \times 10^{-5}$ at all contrasts).

We calculated the approach selectivity index for each cell (*Figure 6H*). Unlike to the moving textures, Off smooth monostratified and broad thorny cells lacked consistent approach selectivity to the annulus stimulus, but approach selectivity persisted in On smooth monostratified and On and Off parasol cells. Besides demonstrating that approach selectivity generalized to a broader range of visual stimuli in the latter three cell types, these results aligned well with the predictions of the coupled subunit model, suggesting that electrical coupling or other lateral interactions within the network contributes to approach selectivity in these cells. To gain further insight into the circuit mechanisms involved, we performed synaptic current recordings from parasol cells to the moving annulus stimulus paradigm.

## Distinct contributions of the On and Off visual pathways to approach motion selectivity

We presented the approaching and receding annuli while recording the excitatory and inhibitory synaptic inputs to parasol ganglion cells (see Materials and methods). Excitatory and inhibitory currents were measured by holding a cell's membrane voltage at the reversal potentials for inhibitory ($E_{Cl}$, –70 mV) and excitatory synaptic currents ($E_{cation}$, 0 mV), respectively. The pattern of excitatory synaptic inputs mirrored the observed spiking pattern—excitatory currents were largest for approaching annuli matching the cell's preferred contrast polarity—positive contrasts in On cells and negative contrasts in Off cells (compare *Figure 6*, *Figure 7*). For example, 100% preferred-contrast approaching annuli evoked excitatory inputs that were much larger than receding annuli in both On (approaching, $40.4 \pm 9.3$ pC; receding, $0.7 \pm 5.5$ pC; n = 16 cells; p=$4.4 \times 10^{-4}$) and Off parasol cells (approaching, $97.5 \pm 17.2$ pC; receding, $29.0 \pm 10.5$ pC; n = 19 cells; p=$1.3 \times 10^{-4}$). In addition, receding annuli that were of a non-preferred contrast evoked larger responses than approaching annuli of the same contrast. These data indicated that the sensitivity to approaching motion observed in the parasol cell spike outputs was present in the excitatory synaptic inputs from diffuse bipolar cells to parasol ganglion cells (*Figure 7*).

Inhibitory synaptic input to parasol cells showed the opposite pattern to that of excitation. Preferred-contrast receding annuli produced larger inhibitory currents than approaching annuli of the same contrast polarity (On: approaching, $89.4 \pm 26.9$ pC; receding, $127.4 \pm 28.6$ pC; n = 13 cells; p=0.15; Off: approaching, $64.1 \pm 27.7$ pC; receding, $114.4 \pm 29.3$ pC; n = 13 cells; p=$1.7 \times 10^{-2}$). This finding indicated that the increased selectivity to approaching motion was mediated by a combination of increased synaptic excitation and reduced synaptic inhibition relative to receding motion; this pattern of synaptic input amplified the differences in parasol cell responses to approaching and receding motion.

Our synaptic current recordings revealed an apparent contribution of inhibitory synaptic input to approach selectivity. Inhibitory input to parasol cells arises primarily from crossover inhibition (*Cafaro and Rieke, 2010*; *Cafaro and Rieke, 2013*). This type of inhibition emerges from amacrine

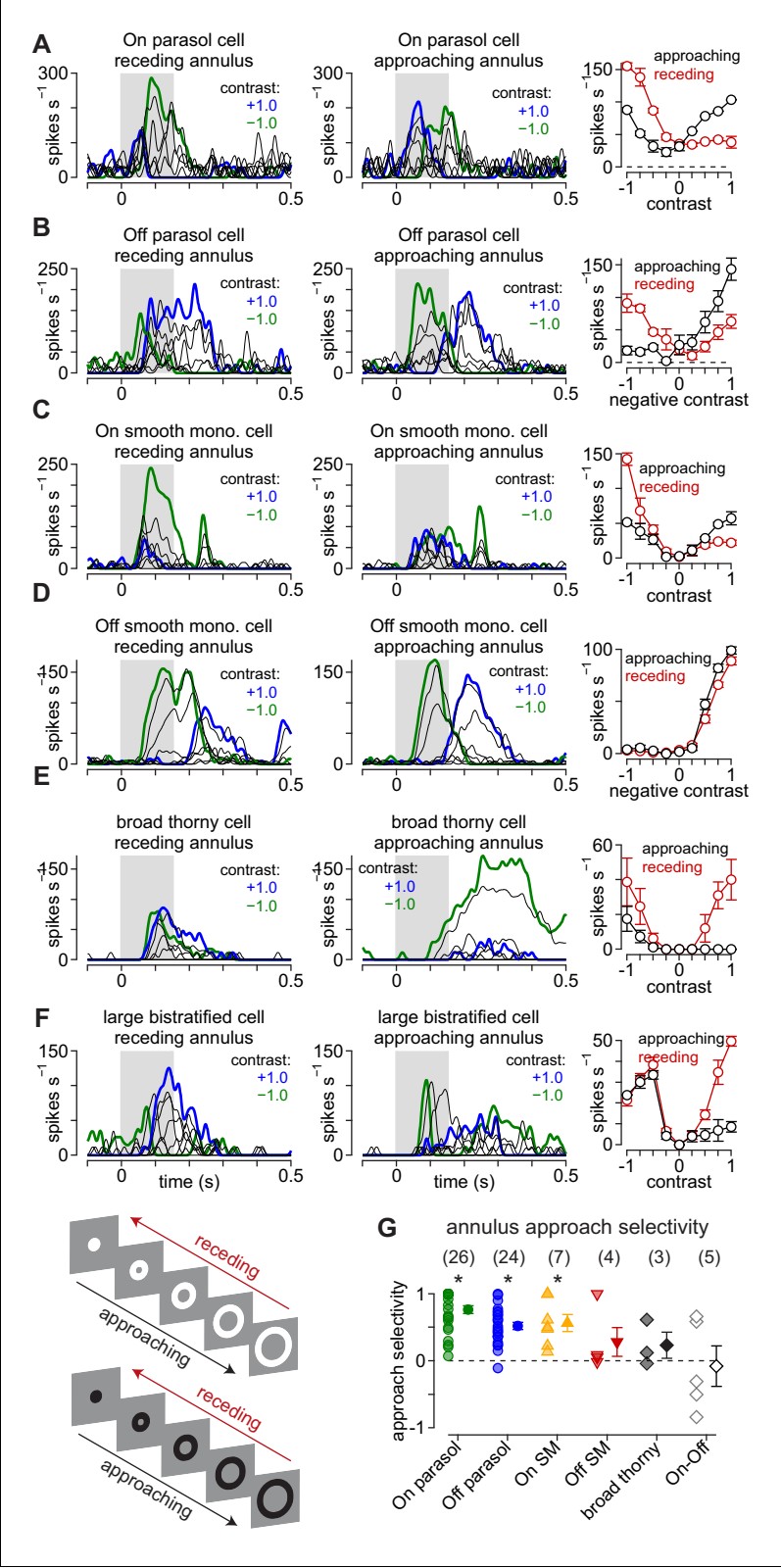

**Figure 6.** Approach motion selectivity for moving annuli. (**A**) Spike responses in an On parasol ganglion cell to receding (*left*) and approaching (*middle*) annuli presented at a series of contrasts. *Right*, Average spike rate during movement of receding (*red*) and approaching (*black*) annuli as a function of stimulus contrast. (**B–F**) Same as (**A**) for Off parasol (**B**), On smooth monostratified (**C**), Off smooth monostratified (**D**), broad thorny (**E**), and large

*Figure 6 continued*

bistratified ganglion cells (F). (G) Approach sensitivity index values for the stimulus paradigm. Transparent shapes indicate individual cells. Opaque shapes and error bars indicate mean ± SEM. Asterisks indicate statistically significant values, determined using the Wilcoxon signed rank test.

The online version of this article includes the following source data and figure supplement(s) for figure 6:

**Source data 1.** Included is a data file containing a structure for the approach selectivity data in annulusspikes.
**Figure supplement 1.** Ganglion cell responses to moving spots.

cells with opposing contrast polarity—Off-type amacrine cells inhibit On parasol cells and On-type amacrine cells inhibit Off parasol cells. In addition to directly inhibiting ganglion cells, circuits for crossover inhibition can modulate glutamate release from retinal bipolar cells by directly inhibiting their synaptic terminals, and this can produce pronounced effects on ganglion cell firing (*Cafaro and Rieke, 2013*; *Liang and Freed, 2010*). Thus, crossover inhibition can manifest in both the direct inhibition onto the ganglion cell and in the excitatory input from bipolar cells.

To measure the presynaptic and postsynaptic contributions of crossover inhibition to approach motion selectivity, we recorded synaptic currents in Off parasol cells while blocking crossover

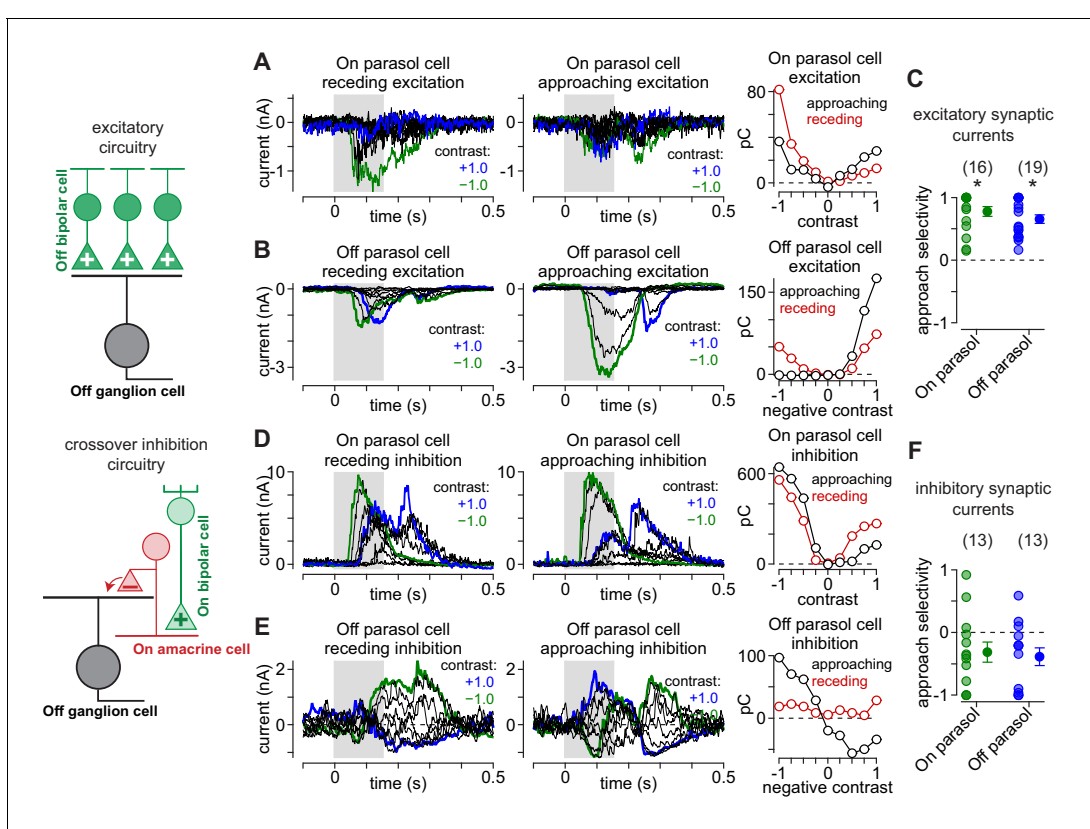

**Figure 7.** Asymmetrical synaptic input patterns underlie approach motion selectivity. (A) Excitatory synaptic currents measured in an On parasol ganglion cell to receding (*left*) and approaching (*middle*) annuli presented at a series of contrasts. *Right*, Excitatory charge during movement of receding (*red*) and approaching (*black*) spots as a function of stimulus contrast. (B) Same as (A) for an Off parasol cell. (C) Approach selectivity index for excitatory synaptic currents for On (*green*) and Off parasol cells (*blue*). Individual cells are shaded; solid circles indicate mean ± SEM. (D) Inhibitory synaptic currents measured in an On parasol ganglion cell to receding (*left*) and approaching (*middle*) annuli presented at a series of contrasts. *Right*, Inhibitory charge as a function of contrast in the On cell. (E) Same as (D) for the Off parasol cell. (F) Approach selectivity index for inhibitory synaptic currents for On (*green*) and Off parasol cells (*blue*). Individual cells are shaded; solid circles indicate mean ± SEM. Statistical significance is indicated with an asterisk and was determined using the Wilcoxon signed rank test.

The online version of this article includes the following source data and figure supplement(s) for figure 7:

**Source data 1.** Included is a data file containing a structure for the approach selectivity data in currents.
**Figure supplement 1.** Crossover inhibition contributes to selectivity for approaching motion.

inhibition with agents that selectively disrupt signaling between photoreceptors and On-type bipolar cells (*Slaughter and Miller, 1981*). We used an mGluR6 agonist/antagonist combination (L-APB, 5 µM; LY341495, 7.5 µM), which has been shown to silence crossover inhibition while minimizing off-target effects in primate retina (*Ala-Laurila et al., 2011*). A comparable pharmacological manipulation was not available for isolating crossover inhibition on On-type ganglion cells (*Manookin et al., 2018*), so we focused on Off parasol cells for these experiments.

Crossover inhibition can act both presynaptically, by modulating bipolar cell glutamate release, and postsynaptically, by directly inhibiting ganglion cell dendrites (*Murphy and Rieke, 2008*; *Pang et al., 2007*; *van Wyk et al., 2009*; *Cafaro and Rieke, 2013*; *Liang and Freed, 2010*; *Molnar et al., 2009*). We measured these effects by recording excitatory and inhibitory synaptic input before and after blocking crossover inhibition (*Figure 7—figure supplement 1*). The effect of blocking crossover inhibition on excitatory synaptic inputs differed for approaching and receding motion. Blocking crossover inhibition did not significantly affect the magnitude of evoked excitatory synaptic currents to approaching motion, but to receding motion, excitatory currents increased following crossover blockade relative to the control condition (*Figure 7—figure supplement 1A, B*). This subsequently reduced approach selectivity relative to the control condition (*Figure 7—figure supplement 1C*; n = 6 cells; p=1.6 × 10$^{-2}$; Wilcoxon signed rank test). These data indicated that during receding motion, the amacrine cell responsible for crossover inhibition provided inhibition at presynaptic bipolar terminals, but this presynaptic inhibition was not present during approaching motion.

These excitatory and inhibitory synaptic recordings indicated that crossover inhibition was more strongly recruited by receding motion than approaching motion. Thus, crossover inhibition enhanced approach motion selectivity by suppressing excitatory synaptic release and directly inhibiting ganglion cell dendrites during receding motion. Despite the apparent contribution of crossover inhibition to approach motion selectivity, excitatory synaptic input was larger for approaching motion than receding motion with crossover inhibition blocked. This indicated that the observed bias for approaching motion was present in both the bipolar cell and amacrine cell circuitries and was amplified by interactions between these circuit elements.

## Discussion

We compared the responses of several primate ganglion cell types to approaching and receding motion. We found that parasol and smooth monostratified ganglion cells consistently showed larger responses to approaching textures and annuli relative to receding stimuli of the same type (*Figure 1*, *Figure 6*). We further demonstrated that the asymmetrical response patterns of these cells to approaching and receding motion arises from the concerted activity of excitatory and inhibitory synaptic inputs to these ganglion cells (*Figure 5*, *Figure 7*, *Figure 7—figure supplement 1*). Further, approach selectivity was weak or absent from several On-Off type cells indicating that it was not a general property of primate ganglion cells. Below, we note some functional implications of these results.

### Contributions of circuit nonlinearities to visual coding

Canonical receptive-field models have been used to describe the visual properties of many ganglion cell types including parasol cells (*Enroth-Cugell and Robson, 1966*; *Chichilnisky and Kalmar, 2002*) and smooth monostratified cells (*Crook et al., 2008*; *Petrusca et al., 2007*). These models can accurately predict neural responses to a small subset of potential visual inputs such as briefly presented stimuli (*Figure 5—figure supplement 1*) or spatiotemporally uncorrelated noise (*Chichilnisky and Kalmar, 2002*), but they perform poorly at predicting responses to stimuli containing spatiotemporal correlations (*Manookin et al., 2018*; *Kuo et al., 2016*), including naturalistic stimuli (*Heitman et al., 2016*; *Turner and Rieke, 2016*; *Turner et al., 2018*). These failures occur because retinal circuits contain many nonlinearities that are not considered in classical receptive field models. Indeed, the traditional linear-nonlinear spatiotemporal model failed to accurately predict the strong asymmetries observed to approaching and receding motion (*Figure 5*, *Figure 5—figure supplement 1*, *Figure 6*).

In vertebrates, the principal retinal nonlinearities arise at the level of bipolar cell glutamate release (*Demb et al., 2001*; *Demb et al., 1999*). Additional nonlinearities are produced by electrical

coupling between bipolar cells (*Kuo et al., 2016*; *Manookin et al., 2018*; *Trenholm et al., 2013*; *Hoggarth et al., 2015*) and by inhibitory amacrine cells (*Kim and Kerschensteiner, 2017*; *Dong and Werblin, 1998*; *de Vries et al., 2011*; *Baccus et al., 2008*). Further, a recent study revealed strong nonlinearities in the interactions between different dendritic branches of smooth monostratified ganglion cells, providing an additional level of nonlinear interactions after the bipolar cell nonlinearities (*Rhoades et al., 2019*). Besides frustrating attempts by researchers to produce accurate computational models of retinal processing, these nonlinearities form the basis for extracting (or rejecting) specific features from visual inputs. For example, amacrine cell input allows the ~15 bipolar cell types to differentially affect the ~30–40 ganglion cell pathways (*Asari and Meister, 2012*; *Asari and Meister, 2014*; *Masland, 2012*; *Wässle, 2004*). Further, these interactions contribute to spatial selectivity (*Greschner et al., 2016*; *Olveczky et al., 2003*; *Cook and McReynolds, 1998*; *Taylor, 1999*; *Flores-Herr et al., 2001*) and short-term plasticity (*Naecker, 2018*; *Kastner and Baccus, 2011*; *Kastner and Baccus, 2013*; *Appleby and Manookin, 2019*; *Kastner et al., 2019*).

The prominent type of inhibitory input at play in approach motion selectivity, crossover inhibition, arises via interactions between the On and Off visual pathways. The circuit motif for crossover inhibition has been conserved across several vertebrate species including fish (*Rosa et al., 2016*), salamanders (*Pang et al., 2007*), guinea pigs (*Liang and Freed, 2010*; *Manookin et al., 2008*), rodents (*Münch et al., 2009*; *van Wyk et al., 2009*), rabbits (*Molnar et al., 2009*), and primates (*Cafaro and Rieke, 2010*; *Cafaro and Rieke, 2013*), and it contributes to a variety of visual functions (reviewed in *Werblin, 2010*). Here, we demonstrate that crossover inhibition amplifies the asymmetrical responses of primate ganglion cells to approaching and receding motion by providing inhibition both at the level of the diffuse bipolar cell terminals and at the level of the ganglion cell dendrites (*Figure 7*, *Figure 7—figure supplement 1*; *Münch et al., 2009*).

In addition, our findings highlight the utility of electrical synapses in neural computation. Electrical coupling between neighboring bipolar cells enhances responses to visual inputs containing spatiotemporal correlations and endows certain ganglion cells with generalized motion selectivity (*Kuo et al., 2016*; *Manookin et al., 2018*) and enhanced direction selectivity in others (*Hoggarth et al., 2015*). Our results here suggest that the effects of electrical synaptic transmission varies with the type of motion—potentiation occurs during approaching motion and was absent during receding motion (*Figure 5*, Appendix 1). Perhaps, the prevalence of these synapses in the retina was partly an adaptation, early in vertebrate evolution, for processing the spatiotemporal correlations that are so common in visual environments (*Völgyi et al., 2013*; *Field, 1987*), including those generated during an animal's own motion through the environment (*Figure 1*, *Figure 4*; *Clifford et al., 1999*; *Schrater et al., 2001*; *Dong and Atick, 1995*).

## Relationship to previous findings

In rodent retina, approach selectivity is found only in a single ganglion cell class—the Off Alpha ganglion cell and is absent from several other types (*Münch et al., 2009*). We found approach selectivity to be more ubiquitous in the primate retina, manifesting in parasol, smooth monostratified, and broad thorny ganglion cells to varying degrees (*Figure 1*, *Figure 6*). This variation may reflect the distinct ethological demands on these species. For example, the need to quickly detect approaching dark objects, such as overhead predators would be necessary for survival in prey species like mice (*Zhang et al., 2012*). Thus, expression of approach selectivity in an Off-type ganglion cell would allow these vital computations to occur early in the visual pathway. Likewise, the arboreal habitats of early primates increased the need to quickly and accurately perform visually guided movements with the arms and hands and led to the expansion of areas in the parietal cortex responsible for these tasks (*Goodale and Milner, 1992*). These habitats may have also produced a need for light/dark symmetry in detecting approaching motion as well as self-motion through the environment (*Clifford et al., 1999*; *Schrater et al., 2001*).

## Contributions to visual processing in primates

The selectivity of these macaque ganglion cell types to approaching motion should be distinguished from the looming-sensitive neurons found in retinorecepient brain regions. For example, looming-sensitive neurons in the optic tectum of pigeons respond when an object moves toward the animal,

but do not respond when the animal moves toward the object (*Sun and Frost, 1998*). The ganglion cells that we tested would not be able to distinguish self-motion and object motion in this way. Instead, the approach motion selectivity that we observed would be an initial step in a series of computations culminating in the detection of approaching objects.

Indeed, the same circuit nonlinearities that enhanced selectivity to approaching motion also produced ambiguities between the direction (approaching/receding) and contrast (light/dark) of moving objects. This result indicates that downstream visual circuits receiving input from parasol and smooth monostratified ganglion cells would also require input from other retinal pathways to resolve these ambiguities. For example, downstream circuits might obtain a more faithful readout of the reflectance of a moving object from the concerted activity of midget ganglion cells and then, with the aid of this information, determine whether the object were approaching or receding. However, future studies will be needed to determine how and where in the visual stream such ambiguities are resolved.

# Materials and methods

Experiments were performed in an in vitro, pigment-epithelium attached preparation of the macaque monkey retina (*Manookin et al., 2015*). Eyes were dissected from terminally anesthetized macaque monkeys of either sex (Macaca *fascicularis*, *mulatta*, and *nemestrina*) obtained through the Tissue Distribution Program of the National Primate Research Center at the University of Washington. All procedures were approved by the University of Washington Institutional Animal Care and Use Committee.

## Tissue preparation and electrophysiology

The retina was continuously superfused with warmed (32–35°C) Ames' medium (Sigma) at ~6–8 mL $min^{-1}$. Recordings were performed from macular, mid-peripheral, or peripheral retina (2–8 mm, 10–30° foveal eccentricity). Physiological data were acquired at 10 kHz using a Multiclamp 700B amplifier (Molecular Devices), Bessel filtered at 3 kHz (8-pole [900 CT, Frequency Devices] in series with 4-pole in Multiclamp), digitized using an ITC-18 analog-digital board (HEKA Instruments), and acquired using the Symphony acquisition software package developed in Fred Rieke's laboratory (http://symphony-das.github.io).

Recordings were performed using borosilicate glass pipettes containing Ames medium for extracellular spike recording or, for whole-cell recording, a cesium-based internal solution containing (in mM): 105 $CsCH_3SO_3$, 10 TEA-Cl, 20 HEPES, 10 EGTA, 2 QX-314, 5 Mg-ATP, and 0.5 Tris-GTP, pH ~7.3 with CsOH, ~280 mOsm. Series resistance (~3–9 MΩ) was compensated online by 50%. The membrane potential was corrected offline for the approximately –10 mV liquid junction potential between the intracellular solution and the extracellular medium. Excitatory and inhibitory synaptic currents were isolated by holding cells at the reversal potentials for inhibitory currents ($E_{chloride}$, ~–70 mV) and excitatory currents ($E_{cation}$, 0 mV), respectively.

## Visual stimuli

Visual stimuli were generated using the Stage software package developed in the Rieke lab (http://stage-vss.github.io) and displayed on a digital light projector (Lightcrafter 4500; Texas Instruments) modified with custom LEDs with peak wavelengths of 405, 505, and 640 nm. Stimuli were focused on the photoreceptor outer segments through a 10X microscope objective. Mean light levels were in the medium photopic regime, (in photoisomerizations [R*] $cone^{-1}$ $s^{-1}$) L-cone: $1.5 \times 10^4$–$1.5 \times 10^5$, M-cone: $1.2 \times 10^4$–$1.3 \times 10^5$, S-cone: $3.9 \times 10^3$–$5.5 \times 10^4$, rod: $3.6 \times 10^4$–$4.0 \times 10^5$. The ratios of L-cone:M-cone:S-cone activations approximate the equal-energy white point after removing the lens (J. Kuchenbecker and M. Manookin, *in preparation*). Contrast values for annuli are given in Weber contrast and for texture stimuli in root-mean-squared (RMS) contrast.

Spike rate values in the text are given relative to the maintained spike rate prior to presenting the stimulus. For several of the stimulus conditions (e.g. receding motion of a preferred contrast), these spike rates fell below zero and, as a result, the values were set to zero when calculating the approach selectivity index.

For extracellular recordings, currents were wavelet filtered to remove slow drift and amplify spikes relative to the noise (*Wiltschko et al., 2008*) and spikes were detected using either a custom

k-means clustering algorithm or by choosing a manual threshold. Spike rate (in spikes s$^{-1}$) was calculated using a Gaussian temporal envelope (SD, 0.67 ms). Prior to analysis, data were downsampled to 1 kHz using a Chebushev filter (type I IIR; filter order, 8). Whole-cell recordings were leak subtracted and responses were measured relative to the median membrane currents immediately preceding stimulus onset (0.25–0.5 s window).

## Stochastic textures

The stochastic texture stimuli used in the model were generated by bandpass filtering a matrix of random noise (*Schrater et al., 2001*). To simulate approaching or receding motion, the center frequency of the filter changed on each frame such that each frame was a rescaled version of the original texture (geometric mean spatial frequency, 1.6 cycles degree$^{-1}$). This bandpass filter was a cosine function in the spatial frequency domain:

$$\hat{F}(\omega) = 0.5 + 0.5\cos(W), -\pi \leq W \leq \pi \tag{2}$$

where

$$W = \log_2\left(\frac{\omega}{2f(t)}\right) \tag{3}$$

where $\omega$ are the spatial frequencies in the image with the $F_0$ component shifted to the center of the spectrum (using the *fftshift* function in MATLAB). Values of $W$ (in radians) were constrained to fall between $\pm\pi$. Texture spatial frequency ($f$) changed exponentially as a function of time:

$$f(t) = exp(\log_e[f_0] - rt) \tag{4}$$

where $f_0$ is the peak frequency of the filter at time zero and $r$ is the rate of texture expansion in Hz. Spatial frequency proceeded from the highest to the lowest values for approaching textures and from the lowest to the highest values for receding textures as in *Equation 4*.

## Difference-of-Gaussians receptive-field model

For each of the computational circuit models, the parasol cell receptive field was modeled as a difference-of-Gaussians. Receptive-field parameters were measured using sinusoidally modulated spots that varied in size. Spike responses were fit with *Equation 5* (*Enroth-Cugell et al., 1983*; *Troy et al., 1999*):

$$R = w_{center}\left(1 - exp\left(-\frac{r^2}{2\sigma^2_{center}}\right)\right) - w_{surround}\left(1 - exp\left(-\frac{r^2}{2\sigma^2_{surround}}\right)\right) \tag{5}$$

where $w_x$ is the weighting of the center or surround and $_x$ is the standard deviation of the center or surround. The sizes and weightings of center and surround regions were then used in the pooling stage of our computational models.

## Determining the difference in kinetics between center and surround

The kinetics of center and surround regions of the receptive field were measured using a Gaussian temporal flicker stimulus. On each stimulus frame, center or surround regions were uniformly presented with a single contrast which was drawn pseudo-randomly from a Gaussian distribution with a mean of 0.0 and a standard deviation of 0.1. Temporal filters were then determined by cross-correlating the presented contrast trajectory ($S$) with the cell's spike output ($R$; *Equation 6*; *Baccus and Meister, 2002*).

$$F(t) = \int R(\tau)S(t+\tau)d\tau \tag{6}$$

These filters were then modeled as a damped oscillator with an S-shaped onset (*Schnapf et al., 1990*; *Angueyra and Rieke, 2013*) as described by *Equation 7*,

$$F(t) = A\frac{(t/\tau_{rise})^n}{1 + (t/\tau_{rise})^n}exp(-t/\tau_{decay})\cos\left(\frac{2\pi t}{\tau_{period}} + \varphi\right) \tag{7}$$

where $A$ is a scaling factor, $\tau_{rise}$ is the rising-phase time constant, $\tau_{decay}$ is the damping time constant, $\tau_{period}$ is the oscillator period, and $\varphi$ is the phase (in degrees). For surround subunits, a temporal lag of 15 ms was included in the temporal component of the receptive field to account for the delay relative to the center (see *Figure 2*).

The relationship between input and output (i.e. the nonlinearity) was calculated by convolving the temporal filter and stimulus to generate the linear prediction ($P$).

$$P(t) = \int F(\tau)S(t-\tau)d\tau \tag{8}$$

The prediction (x-axis) and response (y-axis) were modeled as a cumulative Gaussian distribution (*Chichilnisky, 2001*).

$$N(x) = \varepsilon + \frac{\alpha}{\sqrt{2\pi}}\int_{-\infty}^{x} e^{\frac{-(\beta t+\gamma)^2}{2}}dt \tag{9}$$

where $\alpha$ indicates the maximal output value, $\epsilon$ is the vertical offset, $\beta$ is the sensitivity of the output to the generator signal (input), and $\gamma$ is the maintained input to the cell. In practice, *Equation 9* was invoked using MATLAB's cumulative distribution function (*normcdf*).

## Neural circuit models

We created models of the retinal circuitry to gain a deeper understanding of how synaptic nonlinearities and circuit motifs could shape neural response properties. The model was implemented in the following stages:

1. Space-time filtering stage

   - Generate the subunit spatiotemporal receptive fields ($F$).
   - Generate a stimulus ($S$) with the same dimensionality as the receptive field. Stimulus values are given in contrast.
   - Generate the subunits linear response ($R$) by convolving the stimulus and receptive field; add Poisson noise.
2. Coupling stage (coupling model only)
   - Calculate the Euclidean distance between each pair of subunits ($d$).
   - Calculate the change in current in each subunit due to simulated electrical coupling.
3. Subunit input-output stage

   - Pass the result ($R$) through the appropriate input-output function.
4. Pooling stage

   - Apply temporal delay to subunits forming the ganglion cell's receptive-field surround.
   - Weight each subunit according to its distance from the model ganglion cell's receptive field center.
   - Sum subunit inputs to the model ganglion cell.

### Stage 1: Subunit space-time filtering

We first generated a hexagonal grid of model subunits with an average spacing of 32 μm between neighboring units. The location of each subunit was randomly shifted in the *x* and *y* dimensions to simulate randomness in the bipolar cell mosaic (s. d. ±2 μm). Subunit spatial filtering was modeled with a difference-of-Gaussians receptive-field model (*Equation 5*) using parameters based on previous measurements from diffuse bipolar cells in macaque retina (*Dacey et al., 2000*; *Boycott and Wässle, 1991*; *Tsukamoto and Omi, 2015*; *Tsukamoto and Omi, 2016*). Temporal filtering was performed using parameters from *Equation 7* obtained by direct measurement of excitatory synaptic outputs of diffuse bipolar cells onto parasol cell dendrites (*Manookin et al., 2018*). Thus, the subunit's spatiotemporal receptive field ($F$) was the product of a two-dimensional difference-of-Gaussians (spatial domain, *x*) and a temporal filter (time domain, *t*). The output of each subunit ($R$) was determined by convolving the stimulus ($S$) with the subunit's spatiotemporal receptive field ($F$) as described in *Equation 10*.

$$R(t) = \int_0^t d\tau \int d^2x \, F(\vec{x}, \tau) \, S(\vec{x}, t - \tau) \tag{10}$$

Random fluctuations in membrane potential were simulated by adding Poisson noise to subunit responses; noise values were based on our previous direct measurements of diffuse bipolar cell synaptic outputs (*Manookin et al., 2018*).

## Stage 2: Apply coupling between subunits
For the coupling model, we first calculated the Euclidean distance between model subunits from their x- and y-locations.

$$d_{ij} = \sqrt{(x_i - x_j)^2 + (y_i - y_j)^2} \tag{11}$$

where $d_{ij}$ is the distance between the *i*th and *j*th subunits.

Subunits in the coupling model shared a portion of their output based on differences in driving force and distance between subunits (*Equation 12*). The response of each subunit following coupling was determined by adding the change due to coupling to the response prior to coupling ($R_0$).

$$R_i(t) = R_{0i}(t) + \left[ \sum_{j=1}^n g \left( R_{0i}(t) - R_{0j}(t) \right) exp(-d_{i,j}/\lambda) \right] \tag{12}$$

where *g* is the coupling gain or portion of the response shared between subunits, $\lambda$ is the coupling length constant, $d_{i,j}$ is the pairwise Euclidean distance between the *i*th and *j*th subunits, and *n* is the total number of subunits in the model.

## Stage 3: Subunit input-output functions
The response of each subunit was then passed through the appropriate input-output function— responses in the linear subunit model were passed through a linear function (i.e., $y = x$) and the non-linear and coupled subunit models were passed through the nonlinear function that we directly measured from excitatory synaptic inputs to parasol ganglion cells (see *Equation 9*).

## Stage 4: Pooling
The final stage of the model, the pooling stage, was then performed for each of the three subunit models. Model ganglion cell responses were the weighted ($w_x$) sum of inputs from center and surround subunits ($z_x$).

$$z_{RGC}(t) = w_{center} z_{center}(t) - w_{surround} z_{surround}(t) \tag{13}$$

where the weightings of the center and surround regions of the ganglion cell receptive field ($w_x$) were determined via direct measurements from parasol cells (see *Figure 2*, *Equation 5*). To determine the pooled inputs from center and surround subunits ($z_x$), subunit responses (*N*) following spatiotemporal filtering (linear subunit model) or the output nonlinearity (nonlinear and coupled subunit models) were pooled according to *Equation 14*,

$$z_x(t) = \sum_{i=1}^n N_i(t) \, exp(-d_i^2/2\sigma_x^2) \tag{14}$$

where $d_i$ is the Euclidean distance from the *i*th subunit's receptive field center to the center of the ganglion cell's receptive field and $\sigma_x$ is the standard deviation of the center or surround regions of the ganglion cell's receptive field (see *Figure 2*, *Equation 5*).

## Decoding models
We employed two decoding models to better understand how accurately downstream visual circuits could determine the direction of texture motion from the outputs of model On and Off parasol cells.

The linear model summed the scaled outputs of the model On and Off cells as described in *Equation 15*,

$$f_{\text{LINEAR}} = a_{\text{ON}}r_{\text{ON}} + a_{\text{OFF}}r_{\text{OFF}} \tag{15}$$

where $a_{ON}$ and $a_{OFF}$ are scaling constants. The quadratic model was similar in structure except that the response from each pathway was squared prior to summation (*Equation 16*),

$$f_{\text{QUADRATIC}} = a_{\text{ON1}}r_{\text{ON1}} + a_{\text{ON2}}r_{\text{ON2}}^2 + a_{\text{OFF1}}r_{\text{OFF1}} + a_{\text{OFF2}}r_{\text{OFF2}}^2 \tag{16}$$

We evaluated the ability of the decoding models to distinguish between approaching and receding textures using the Jensen-Shannon distance. Model output was discretized by rounding to the nearest integer value. The Jensen-Shannon distance (i.e. the square-root of the Jensen-Shannon divergence) was calculated from the Kullback-Leibler divergence ($D_{KL}$) between the probability distributions for the model outputs to each approaching ($P$) and receding ($Q$) texture sequence.

$$JS_{dist}(P,Q) = \sqrt{\frac{1}{2}\left[D_{KL}(P\frac{P+Q}{2}) + D_{KL}(Q\frac{P+Q}{2})\right]} \tag{17}$$

The Kullback-Leibler divergence between the model output distributions was calculated according to *Equation 18*.

$$D_{KL}(PQ) = \sum_n p_n \log_2\left(\frac{p_n}{q_n}\right) \tag{18}$$

where $p_n$ is the probability of observing an output of magnitude $n$ in the sample window during approaching motion and $q_n$ is the probability of observing an output of $n$ in the sample window during receding motion.

## Quantification and statistical analysis

All statistical analyses were performed in MATLAB (R2018b, Mathworks). Reported p values in this study were paired and were calculated using the Wilcoxon signed-rank test. Final figures were created in MATLAB (version R2018b), Igor Pro (version 8), and Adobe Illustrator.

## Acknowledgements

We thank Shellee Cunnington, Mark Cafaro, and Jim Kuchenbecker for technical assistance. Tissue was provided by the Tissue Distribution Program at the Washington National Primate Research Center (WaNPRC; supported through NIH grant P51 OD-010425), and we thank the WaNPRC staff, particularly Chris English and Audrey Baldessari, for making these experiments possible. Fred Rieke and Chris Chen assisted in tissue preparation. We thank Fred Rieke for helpful discussions. We also thank Ione Fine, Phil Mardoum, Christian Puller, Greg Schwartz, and Peter Sterling for feedback on a previous version of this manuscript. This work was supported in part by grants from the NIH (NEI R01-EY027323 to MBM; NEI R01-EY029247 to EJC, FR, and MBM.; NEI P30-EY001730 to the Vision Core), Research to Prevent Blindness Unrestricted Grant (to the University of Washington Department of Ophthalmology), Latham Vision Research Innovation Award (to MBM), and the Alcon Young Investigator Award (to MBM).

## Additional information

### Funding

| Funder | Grant reference number | Author |
| --- | --- | --- |
| National Eye Institute | R01-EY027323 | Michael B Manookin |
| National Eye Institute | R01-EY029247 | Michael B Manookin |
| National Eye Institute | P30-EY001730 | Michael B Manookin |

| National Institutes of Health | P51 OD-010425 | Michael B Manookin |
|---|---|---|
| Research to Prevent Blindness | | Michael B Manookin |
| Alcon Research Institute | Alcon Young Investigator Award | Michael B Manookin |
| University of Washington | Latham Vision Research Innovation Award | Michael B Manookin |

The funders had no role in study design, data collection and interpretation, or the decision to submit the work for publication.

## Author contributions
Todd R Appleby, Data curation, Investigation; Michael B Manookin, Conceptualization, Resources, Data curation, Software, Formal analysis, Supervision, Funding acquisition, Validation, Investigation, Visualization, Methodology, Project administration

## Author ORCIDs
Michael B Manookin (ID) https://orcid.org/0000-0001-8116-7619

## Ethics
Animal experimentation: All procedures were approved by the University of Washington Institutional Animal Care and Use Committee (IACUC protocol #4277-01).

## Decision letter and Author response
Decision letter https://doi.org/10.7554/eLife.51144.sa1
Author response https://doi.org/10.7554/eLife.51144.sa2

## Additional files
### Supplementary files
• Transparent reporting form

### Data availability
We have made the population data in the study freely available. Source data files have been provided for Figures 1, 6, and 7.

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

## Appendix 1

# Organization of the subunit network and shape of the input-output curve confers activation bias

We sought insight into the mechanisms that biased electrically coupled networks to approaching motion. To do this, we created a simplified model of subunit interactions. During approaching motion, activation through electrical coupling spreads from a few subunits at the very center of the ganglion cell receptive field to a greater number of subunits located in more distal regions of the receptive-field center. Likewise, our model contained two types of subunits: proximal subunits that were analogous to the more centrally located bipolar cells and distal subunits that were analogous to the bipolar cells located farther out in the receptive-field center.

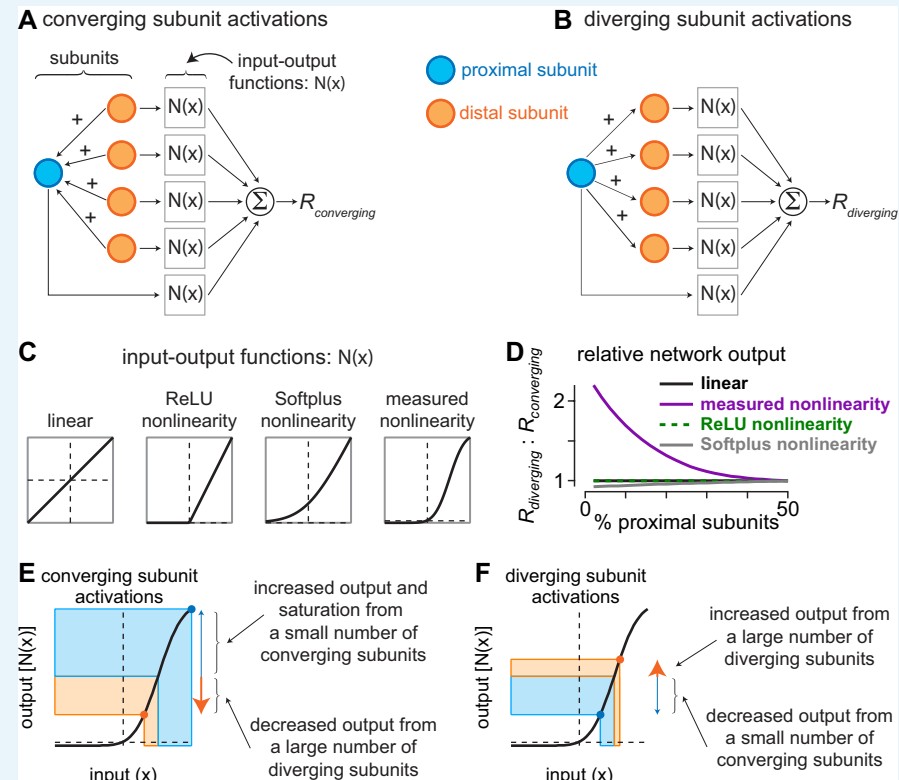

**Appendix 1—figure 1.** Current redistribution and nonlinearity shape determine network bias. (**A**) Activation pattern for converging subunit network. Distal subunits (*orange*) imparted a portion of their input to proximal subunits (*blue*). Subunit inputs then pass through input-output functions prior to being summed as the converging response ($R_{converging}$). (**B**) Activation pattern for diverging subunit network in which proximal subunits impart a portion of their current to distal subunits. (**C**) Shapes of different input-output functions evaluated with the network model. (**D**) Ratio of outputs from diverging ($R_{diverging}$) and converging ($R_{converging}$) activations (*y*-axis) as a function of the percentage of proximal subunits in the network (*x*-axis). The nonlinearity measured directly from excitatory synaptic inputs to parasol cells produced a large bias for diverging network activation when distal subunits outnumbered proximal subunits by 3:1. (**E–F**) Contributions of nonlinearity shape to output bias of subunit networks. (**E**) During convergent activation, the numerical mismatch between proximal and distal subunits and the shape of the input-output nonlinearity drives the small number of proximal subunits to saturating regions of the output curve, whereas the large number of distal subunits decrease their outputs. This results in a relatively small network output ($R_{converging}$). (**F**) During

divergent activation, a small number of distal subunits are suppressed while a large number of distal subunits are potentiated, causing the network output to increase ($R_{diverging}$).

We simulated the subunit activation patterns encountered during approaching and receding annuli with diverging and converging network activations, respectively. Diverging network activation occurred when proximal subunits imparted current to the distal subunits, and converging activation occurred when the distal subunits imparted current to proximal subunits. In all cases, the total amount of input current in the network was conserved (x-axis). Following subunit activation, the input of each subunit was passed through an input-output function. Four different input-output functions were tested—a function in which the mapping between input and output was linear (i.e., $y = x$) and three nonlinear functions with differing shapes to their input-output relationships.

To determine whether the overall output of the network differs with network configuration, we varied the ratio of proximal and distal subunits and we computed the ratio of network output for diverging activation ($R_{diverging}$) and converging activation ($R_{converging}$). For the linear and rectified linear unit (ReLU) input-output functions, the ratio of diverging and converging network activations was equivalent for all conditions tested, and the Softplus nonlinearity also lacked a bias for diverging subunit activation (Selectivity to approaching motion in retinal inputs to the dorsal visual pathway *Appendix 1—figure 1D*). However, the input-output function that we directly measured in parasol ganglion cells produced a very different result. Under conditions in which the relative number of proximal subunits was low, the network output was much (>100%) larger for diverging network activation than for converging activation, reminiscent of the approach motion bias observed in our subunit models (*Figure 3*, *Figure 5*).

Why does the shape of the input-output function produce such different network behaviors? The key to answering this question is in understanding the way in which the input currents are redistributed during network activation. When subunits impart a portion of their current, this current is divided equally among the recipient subunits. Thus, the current from a large number of subunits would produce a large positive movement along the input (x) axis if it were divided among a relatively small number of recipient subunits, and the subunits imparting the current would then show smaller negative movements along the input axis. Under conditions in which the movements occur along linear regions of the input-output function, the loss of current by imparting subunits and the gain in current by recipient subunits would cancel. However, if this gain in current occurred in a region in which the function saturates, as is the case for the function we directly measured in parasol cells, further increases along the input axis would produce negligible changes along the output axis; however, loss of current by imparting subunits would still produce a decrease along the output axis, resulting in a relatively small network output (Selectivity to approaching motion in retinal inputs to the dorsal visual pathway *Appendix 1—figure 1E*). Indeed, this is precisely the case in our network model when distal subunits greatly outnumber proximal subunits—converging network activation increases the inputs of a relatively small number of proximal subunits and decreases inputs of a large number of distal subunits. Proximal subunits are then pushed to regions of the curve that produce saturation in the output. The output of distal subunits, however, decreases and this, coupled with the saturation of proximal subunits, produces a relatively small output from the network.

Diverging subunit activation produces a very different pattern. A relatively small number of proximal subunits impart a portion of their current to distal subunits. As a result, the proximal subunits decrease their outputs, while the distal subunits increase their outputs. Because the imparted current of a few proximal subunits is spread out over many more distal subunits, these distal subunits are unlikely to end up in a saturating region of the curve. Together, these factors result in a relatively large network output (Selectivity to approaching motion in retinal inputs to the dorsal visual pathway *Appendix 1—figure 1F*).

These simplified network simulations highlight two principles that are key to understanding how bias for approaching motion can arise from networks of electrically coupled subunits. First, the sequence in which the subunits are activated is critical—activation of a few subunits that then spreads to a larger number of subunits, as is the case during approaching motion,

potentially produces the largest network outputs (Selectivity to approaching motion in retinal inputs to the dorsal visual pathway *Appendix 1—figure 1D*). Second, the shape of the input-output function is essential to the observed effects. Linear (linear; ReLU) or accelerating (e.g., Softplus) functions did not strongly affect network output. The nonlinearity we directly measured with our synaptic input recordings was sigmoidal, causing saturation at large positive values along the input axis. This saturation favors conditions in which activation is distributed over a larger number of subunits rather than strongly activating a few subunits. Thus, prudent selection of the shape of an input-output nonlinearity can be as important to neural circuit function as the placement of that nonlinearity within the circuit.

