## [Decision Letter]

**Acceptance summary:**

This paper points to an intriguing aspect of visual processing in the macaque retina: Many retinal ganglion cells have an intrinsic bias for approaching objects over receding objects. Even more intriguing is the fact that the current standard model of retinal processing largely predicts the effects correctly.

**Decision letter after peer review:**

Thank you for submitting your article "Motion produces ambiguities in the retinal code for contrast" for consideration by *eLife*. Your article has been reviewed by three peer reviewers, including Markus Meister as the Reviewing Editor and Reviewer #1, and the evaluation has been overseen by Joshua Gold as the Senior Editor.

The reviewers have discussed the reviews with one another and the Reviewing Editor has drafted this decision to help you prepare a revised submission.

Summary:

This is an interesting report on visual processing in the macaque retina. It includes electrical recordings from 5 types of retinal ganglion cell under a broad range of visual stimuli. This is supported by computational modeling to evaluate the potential circuit mechanisms underlying the responses. The focus is on stimuli that include some form of motion. Although these ganglion cells are not generally discussed as representing image motion, they produce some seemingly paradoxical responses. In particular, several of the types have a strong bias in favor of approaching motion over receding motion. These are intriguing observations and suggest that certain visual features useful to higher visual processing are already precomputed in the retina. However, several major concerns were raised that would need to be addressed in a revised submission.

Essential revisions:

1) Many of your results, in particular those from experiments using flashing/expanding/contracting spots and annuli (Figure 1, Figure 2, Figure 3 and Figure 4), were thought to be fully expected from current understanding of retinal processing. Both expanding bright spots and contracting dark spots lead to an overall increase in luminance over the receptive field. Conversely both expanding dark spots and contracting bright spots lead to an overall decrease in luminance. Obviously, one expects that the ON parasols respond preferentially to brightening over dimming. The responses to annuli (Figure 4) can be understood as well: each annulus has an advancing On edge and an advancing Off edge. But the outer edge is much longer, and therefore crosses more receptive field subunits per unit time than the inner edge. This simple observation accounts – at least qualitatively – for all the reported asymmetries in Figure 1, Figure 2, Figure 3 and Figure 4. Thus, the paper should be revised to clarify what, exactly, is expected from previous findings and what is novel.

2) The texture results were thought to be more interesting, but even in that case it seems possible that standard models can explain the effects: At a small scale, the black and white texture elements within a receptive field subunit cancel. As the elements grow, they fill entire subunits, and those subunits that are excited pass that on to the ganglion cell via the rectifying synapse. This scheme could produce your results, but responses generated by expanding/contracting stimuli will also depend on the spatial frequency tuning of the neuron, that is the center-surround interactions in the receptive field. At a scale where the texture elements invade the surround of the subunits one can imagine the bias reversing towards contraction. Do you have data/analyses to illuminate the range of conditions that produce a bias towards expansion or contraction?

3) The standard model of retinal processing that you use to test these null hypotheses was considered inadequate, both in terms of its design and its description. Some details are present in Figure 2C but it probably merits a full figure. Two models in particular should be considered: (A) An RGC with a linear receptive field with an excitatory center and an antagonistic surround, each with a biphasic time course that roughly matches the cells studied. (B) A model with nonlinear subunits. In this model, the receptive field center is composed of subunits (bipolar cells) that perform linear integration over their spatio-temporal center-surround receptive field. The output of each subunit is rectified (synaptic transmission). Then the RGC pools over these subunits. The RGC may pool over an additional antagonistic surround (wide-field amacrine cells). Because temporal processing is essential for the phenomena discussed here, the time course of all these components should be reasonable compared to what is known. This picture of generic retinal processing emerged from the literature of the past 20 years or so, and has proven very successful in explaining many phenomena.

4) There was also general agreement that the writing and focus could be improved substantially. The manuscript in its present form buries the interesting message under an excess of material (~150 figure panels). In general, the novel results should be emphasized, particularly those using the texture stimuli. It would also be useful to frame those findings in the context of other, related results regarding human perception of expanding patterns (Schrater, 2001). That paper has been followed up with reports that similar selectivity for expanding textures exists among neurons in V1 and already in LGN (e.g. Wang and Yao, 2011 and doi: https://doi.org/10.1167/11.3.5). The current manuscript places the origins already in the retina, which if true would be novel and interesting.

5) Reverse correlation to annuli: There was consensus that the analysis of Figure 6 did not add further insight. Also, reviewers questioned whether the assumptions underlying this application of reverse correlation were valid at all: Reverse correlation produces rational results if the system can be treated as a generalized linear model. That means the stimulus variable should be weighted linearly before some nonlinear transformation. So a stimulus of radius 100 should contribute twice as much as a stimulus of radius 50. We know that is not the case. Large enough annuli will enter the receptive field surround and get negative weights.

6) Circuit interpretations:

Electrical coupling: the model of Figure 9 is speculative and should be omitted or given more detail. Is this simply a way to make larger subunits prior to rectification? Or something more subtle?

Inhibition: Include consideration of feedback and lateral inhibition in the circuit models. Both could play a major role in the effects, in addition to crossover inhibition. Pharmacology (e.g. GABAc block, glycine block) experiments could resolve these contributions.

7) The discussion of "ambiguity" is confusing and should be omitted, starting with the title. The response of any single neuron, like an RGC, always leaves complete ambiguity about the stimulus that caused it. This is because the RGC only reports a single function of time (its firing rate) whereas the stimulus has an infinity of pixels that can vary in time. So one can construct infinitely many stimuli that will produce the identical response. Why should one choose among that huge iso-response set some arbitrary pair of stimuli, like approaching/receding annuli of opposite contrast, and then claim that the ambiguity is about contrast? A complete statement of what the neuron encodes is given by its response model, and as argued above one can understand the claimed "ambiguities" based on the standard response model in which On cells respond to local brightening and Off cells to local dimming.

[Editors' note: further revisions were suggested prior to acceptance, as described below.]

Thank you for resubmitting your work entitled "Selectivity to approaching motion in retinal inputs to the dorsal visual pathway" for further consideration by *eLife*. Your revised article has been evaluated by Joshua Gold as Senior Editor and a Reviewing Editor.

The manuscript has been improved but there are some remaining issues that need to be addressed before acceptance, as outlined below:

Essential revisions:

The range of texture stimuli: As discussed in the first review one might expected that when texture stimuli become too coarse (covering the RGC antagonistic surround), receding motion is preferred. The report states that stimuli begin at 0.8 cycles / degree, the approximate width of the ganglion cell dendritic tree. Are there results from lower spatial frequencies, where one might expect the bias to reverse? If not, please test this range in the model and mention in the Results section.

The coupled subunit model: More detail is needed here, for example the assumed gain and length constant of electrical coupling. How do these assumed parameters relate to known properties of electrical coupling? Is the computed approach bias robust over a range of parameters? Also, given that the electrical coupling occurs before the bipolar output nonlinearity (subsection “Approach motion selectivity present for moving annuli”), this is formally equivalent to assuming a larger receptive field for the bipolar cell. How certain are we about the size of these receptive fields, and how much were they enlarged by your assumed coupling? Clarify in the Materials and methods section and Results section.

The nonlinear center-surround subunit model: It still is not clear why this model fails to reproduce the approach bias for annuli (Figure 5C). For approaching motion, the center will be activated before the surround. In this case the inhibition from wide-field amacrine cells to bipolar cells will be delayed and not affect release. For receding motion, the surround will be activated prior to center stimulation, allowing inhibition to take hold and dampen glutamate release. As mentioned above, the model with electrical coupling is equivalent to one without coupling but with larger BC receptive fields. Why does that make such a difference (Figure 5D)? Additional interpretation of these results will help the reader understand them.

The Materials and methods section on receptive field models is hard to understand (tested by one of our PhD students who was given the assignment to replicate the models). For example, what is the subunit's spatio-temporal receptive field in Equation 10. It would be useful if you took the reader through each of the models from start (photons) to finish (spikes) with one continuous series of equations.

---

## [Author Response]

Summary:This is an interesting report on visual processing in the macaque retina. It includes electrical recordings from 5 types of retinal ganglion cell under a broad range of visual stimuli. This is supported by computational modeling to evaluate the potential circuit mechanisms underlying the responses. The focus is on stimuli that include some form of motion. Although these ganglion cells are not generally discussed as representing image motion, they produce some seemingly paradoxical responses. In particular, several of the types have a strong bias in favor of approaching motion over receding motion. These are intriguing observations and suggest that certain visual features useful to higher visual processing are already precomputed in the retina. However, several major concerns were raised that would need to be addressed in a revised submission.

We thank the reviewers for their careful reading of our manuscript and their helpful feedback. In accordance with the reviewers’ input, we have performed additional experiments and have edited the text and figures to increase clarity and readability. We feel that this work has significantly improved the manuscript. Below, we address the points raised by the reviewers.

Essential revisions:1) Many of your results, in particular those from experiments using flashing/expanding/contracting spots and annuli (Figure 1, Figure 2, Figure 3 and Figure 4), were thought to be fully expected from current understanding of retinal processing. Both expanding bright spots and contracting dark spots lead to an overall increase in luminance over the receptive field. Conversely both expanding dark spots and contracting bright spots lead to an overall decrease in luminance. Obviously, one expects that the ON parasols respond preferentially to brightening over dimming. The responses to annuli (Figure 4) can be understood as well: each annulus has an advancing On edge and an advancing Off edge. But the outer edge is much longer, and therefore crosses more receptive field subunits per unit time than the inner edge. This simple observation accounts – at least qualitatively – for all the reported asymmetries in Figure 1, Figure 2, Figure 3 and Figure 4. Thus, the paper should be revised to clarify what, exactly, is expected from previous findings and what is novel.

Thank you. We agree with the reviewers’ that the moving spot results were expected. We made this same observations in the original text and used it as motivation for using the annulus stimulus paradigm. The stimulus was included originally to demonstrate the original point that certain combinations of spot contrast and approaching/receding motion could elicit the same response output in these cells. We have moved this figure to the figure supplements.

While we agree that the response patterns observed to the spot stimulus paradigm were expected, we disagree with the premise that the annulus responses were also expected. In the original study, we used direct recordings and two subunit models (see comment #3 below) to illustrate that the annulus stimulus paradigm should not produce large differences in responses to approaching and receding motion (see Figure 2 and Figure 3 in the original work and Figure 5 of the current work). If the receptive-fields of the recorded ganglion cells were space-time invariant (as is popularly believed) approach selectivity would be absent for the moving annulus paradigm. Only when the receptive fields contain components that bias the cells’ responses to motion, do we expect to see approach selectivity. To address the reviewers’ concern, we have reorganized the figures and updated the text. We have expanded the modeling figures and now clarify the predictions of different circuit motifs to these types of motion (see current Figure 5).

2) The texture results were thought to be more interesting, but even in that case it seems possible that standard models can explain the effects: At a small scale, the black and white texture elements within a receptive field subunit cancel. As the elements grow, they fill entire subunits, and those subunits that are excited pass that on to the ganglion cell via the rectifying synapse. This scheme could produce your results, but responses generated by expanding/contracting stimuli will also depend on the spatial frequency tuning of the neuron, that is the center-surround interactions in the receptive field. At a scale where the texture elements invade the surround of the subunits one can imagine the bias reversing towards contraction. Do you have data/analyses to illuminate the range of conditions that produce a bias towards expansion or contraction?

We thank the reviewers for bringing up this interesting hypothesis. Before explaining how we addressed this question, we want to clarify a potential misunderstanding about the stimulus used in the original manuscript submission. First, the spatial frequencies used in the present study were, by design, in the same range as those used by the original psychophysical study showing approach selectivity in human perception (Schrater et al., 2001). Second, for the original texture experiments the highest peak spatial frequencies used were on the same scale as the dendritic tree diameters of diffuse bipolar cells in the mid-peripheral retina (3.3 cycles/degree). Thus, for these original experiments, we did not expect to see a pronounced effect from the putative mechanism described in the reviewers’ hypothesis. Nonetheless, we sought to directly address the posited idea with new experiments.

We performed additional recordings from parasol (n = 16) and smooth monostratified (n = 7) ganglion cells to a modified texture stimulus protocol. Each cell was presented with approaching and receding texture sequences as before, but the spatial frequency ranges of the textures were varied—four ranges were tested. The highest peak spatial frequencies (i.e., smallest texture scales) were between the approximate dendritic field diameters of diffuse bipolar cells and parasol ganglion cells in the mid-peripheral macaque retina (0.8-3.3 cycles/degree).

The reviewers’ hypothesis predicts that decreasing the spatial frequency content of the texture sequences in this way should diminish or abolish approach selectivity in these cells. Our direct recordings, however, showed the opposite pattern. Varying the scale ranges of the texture sequences did not change the preference for approaching textures. Approach selectivity persisted in parasol and smooth monostratified cells across these stimulus conditions. In fact, approach selectivity increased with increasing texture scale (i.e. decreasing spatial frequency) in both parasol and smooth monostratified cells (Figure 1D). These results indicated that the mechanisms mediating approach selectivity in these cells operate across a wide range of spatial scales. Further, the results showed the opposite pattern than what would be predicted from the reviewers’ proposed hypothesis.

3) The standard model of retinal processing that you use to test these null hypotheses was considered inadequate, both in terms of its design and its description. Some details are present in Figure 2C but it probably merits a full figure. Two models in particular should be considered: (A) An RGC with a linear receptive field with an excitatory center and an antagonistic surround, each with a biphasic time course that roughly matches the cells studied. (B) A model with nonlinear subunits. In this model, the receptive field center is composed of subunits (bipolar cells) that perform linear integration over their spatio-temporal center-surround receptive field. The output of each subunit is rectified (synaptic transmission). Then the RGC pools over these subunits. The RGC may pool over an additional antagonistic surround (wide-field amacrine cells). Because temporal processing is essential for the phenomena discussed here, the time course of all these components should be reasonable compared to what is known. This picture of generic retinal processing emerged from the literature of the past 20 years or so, and has proven very successful in explaining many phenomena.

We are surprised by this criticism from the reviewers. Each of the inadequacies of our model cited by the reviewers in the above paragraph was fully addressed in the original manuscript. The following bullet points outline several figures and areas of the original text which explicitly described the subunit models in the original manuscript and addressed the reviewers’ concerns stated in the above paragraph.

All of the models in the original manuscript were subunit models (Figure 2, Figure 3, Figure 9). This was stated explicitly in the accompanying text, figure legends, and Material and methods section.

All of the models included lateral inhibition with a temporal delay from the receptive-field surround. This temporal delay was not only realistic, it was directly measured in the original experiments (Figure 3A, B).

The subunit filters and nonlinearities were also directly measured with synaptic current recordings (Results section; Materials and methods section).

The sizes and strengths of the receptive-field center and surround components were directly measured for both On and Off cells (Figure 2A, B).

Full descriptions of these subunit models were given in the Results section, the Materials and methods section, and in the relevant figure legends (Figure 2, Figure 3, Figure 9).

The relevant figures also contained diagrams showing the subunits, their input-output nonlinearities, and their outputs being combined in the pooling and normalization stages of the models (Figure 2, Figure 3, Figure 9).

Further, the second subsection of the original Results section began with the phrase, “Subunit models predict…”, also indicating that the models discussed were subunit models.

In an attempt to increase the clarity of our modeling approaches, we have now dedicated more space in both the Results section and Materials and methods section to this aspect of our study. We have also expanded the modeling figure and have attempted to clarify the model structures using clearer diagrams and using longer and more careful descriptions in the Results section and Materials and methods section. We hope the reviewers will find this has improved the readability of the manuscript as well as diminished the chance of causing confusion to the reader.

4) There was also general agreement that the writing and focus could be improved substantially. The manuscript in its present form buries the interesting message under an excess of material (~150 figure panels). In general, the novel results should be emphasized, particularly those using the texture stimuli. It would also be useful to frame those findings in the context of other, related results regarding human perception of expanding patterns (Schrater, 2001). That paper has been followed up with reports that similar selectivity for expanding textures exists among neurons in V1 and already in LGN (e.g. Wang and Yang, 2011 and doi: https://doi.org/10.1167/11.3.5). The current manuscript places the origins already in the retina, which if true would be novel and interesting.

We have rewritten nearly all of manuscript to simplify and clarify our study. Further, we have moved three of the figures from the main text to the supplement.

5) Reverse correlation to annuli: There was consensus that the analysis of Figure 6 did not add further insight. Also, reviewers questioned whether the assumptions underlying this application of reverse correlation were valid at all: Reverse correlation produces rational results if the system can be treated as a generalized linear model. That means the stimulus variable should be weighted linearly before some nonlinear transformation. So a stimulus of radius 100 should contribute twice as much as a stimulus of radius 50. We know that is not the case. Large enough annuli will enter the receptive field surround and get negative weights.

It is unfortunate that the reviewers did not feel that these experiments added to our study. These experiments were intended to measure the speed tuning properties of the observed approach motion selectivity and, simultaneously, provide another test of whether the cells exhibited approach selectivity. We feel that both of these goals were achieved.

While we respectfully disagree with the reviewers’ assessment of these experiments, we have removed the accompanying text from the paper and have moved the figure to the supplementary material. Having stated this, there are several statements made concerning the stimulus paradigm and the accompanying analysis that were in error and we would like to address some of these issues briefly.

1) The idea that a stimulus with a radius of 100 would contribute twice as much as that of a radius of 50 is not correct. The stimulus was an annulus, just as that used in the modeling (Figure 5) and direct recordings (Figure 6, Figure 7). The difference was that, instead of continuous motion to simulate approaching or receding objects, the noise annulus randomly increased or decreased in size on each frame presentation.

2) Reverse correlation analysis. Several previous studies have employed similar noise paradigms to measure spatiotemporal tilt in visual neurons (e.g., (Conway and Livingstone, 2003; Pack et al., 2006; Priebe et al., 2006)). Unlike previous approaches however, our stimulus paradigm was specifically designed to measure bias for centripetal/centrifugal motion; and we felt that this was an appropriate use of the stimulus and analysis paradigm.

3) Regarding the notion of “linear weighting” prior to the output nonlinearity. In theory the reviewers’ statement is absolutely correct. In practice, however, nearly all studies using noise stimuli and reverse correlation analysis (or related analyses) do so in recordings from cells with significant nonlinearities in the upstream circuitry. In other words, the stimulus weightings are very much not linear prior to the static output nonlinearity. This is obviously true of any study employing noise stimuli and reverse correlation to examine visual processing in the cortex, but it is also true of studies that use the technique to measure visual properties of retinal ganglion cells given the strong nonlinearities present in the bipolar cell output and in other components of the retinal circuitry (including the cone photoreceptors). Of course, it would be incredibly tenuous if we used noise stimuli and the accompanying analysis as the sole method for measuring approach selectivity. This was not our intent. Instead, we intended to use it as a method complementing other experimental and analytical paradigms employed in our study.

6) Circuit interpretations:Electrical coupling: the model of Figure 9 is speculative and should be omitted or given more detail. Is this simply a way to make larger subunits prior to rectification? Or something more subtle?Inhibition: Include consideration of feedback and lateral inhibition in the circuit models. Both could play a major role in the effects, in addition to crossover inhibition. Pharmacology (e.g. GABAc block, glycine block) experiments could resolve these contributions.

Thank you. We did not adequately communicate our reasoning behind the premise that electrical coupling between bipolar cells contributed to approach motion selectivity. Part of the challenge in communicating these ideas clearly was that the linear and nonlinear subunit model predictions were separated from predictions of the coupled subunit models in the original text (Figure 3 and Figure 9 in the original work). Thus, to more clearly communicate the different predictions of the underlying circuit motifs, we have combined these three models into a single figure (current Figure 5) and have carefully rewritten the accompanying text to improve clarity. Also, given the reviewers’ concerns that the contribution of coupling is speculative, we have removed potentially speculative statements from the Results section and Discussion section.

Further, the computational models all contain lateral inhibition from the receptive-field surround for both the model bipolar cells subunits and the ganglion cells, yet only the model that included coupling between the bipolar cells subunits reproduced the observed approach selectivity. The mechanism that we think is at play here is described in greater detail in our previous work (Manookin et al., 2018), but we summarize it here briefly for the reviewers.

Figure 8 from Manookin et al., 2018 illustrates our working model for how electrical coupling between diffuse bipolar cells produces selectivity to visual motion in parasol ganglion cells. During visual motion, coupled bipolar cells are stimulated in sequence and a portion of the current in the first bipolar cell in the stimulus sequence travels laterally through gap junctions to the second bipolar cell. This current depolarizes the second bipolar cell by some amount (10% in the example) and, because of the nonlinear relationship between bipolar cell voltage and synaptic output, glutamate release is enhanced in the second bipolar cell (>20% in the example). Thus, electrical coupling biases the output of the bipolar cell network to spatiotemporally correlated stimuli, such as moving objects.

In the case of approaching and receding motion, the leading edge of the annulus activates an increasing number of subunits during approaching motion. The increasing number of subunits is offset by the Gaussian profile of the receptive field for the linear and nonlinear subunit models. For the coupled subunit model, however, the potentiation produced by the expanding annulus is not offset by the receptive-field profile (Figure 5). We have added a supplementary movie showing the activation of each subunit as a function of time during an iteration of each of the models. This movie illustrates the difference between the coupled subunit model and other models nicely (see Figure 5—video 1).

7) The discussion of "ambiguity" is confusing and should be omitted, starting with the title. The response of any single neuron, like an RGC, always leaves complete ambiguity about the stimulus that caused it. This is because the RGC only reports a single function of time (its firing rate) whereas the stimulus has an infinity of pixels that can vary in time. So one can construct infinitely many stimuli that will produce the identical response. Why should one choose among that huge iso-response set some arbitrary pair of stimuli, like approaching/receding annuli of opposite contrast, and then claim that the ambiguity is about contrast? A complete statement of what the neuron encodes is given by its response model, and as argued above one can understand the claimed "ambiguities" based on the standard response model in which On cells respond to local brightening and Off cells to local dimming.

We regret our choice of wording. As the reviewers are well aware, much of the literature assumes that parasol ganglion cells principally or exclusively contribute to our vision by encoding spatial contrast to retinorecipient regions of the brain (Kaplan and Shapley, 1986). This continues to be the prevailing view despite the overwhelming behavioral evidence to the contrary (Merigan and Maunsell, 1990; Schiller et al., 1990a, 1990b). Thus, in the context of the canonical theory a downstream neuron listening to the output of On and Off parasol cells would not be able to determine the contrast of a moving object depending on its trajectory. This was the motivation behind our use of the word and the general organization of the original work.

Given the confusion that this word caused, we have removed it and the surrounding discussion from the text. We have instead framed the findings of our experiments in the context of general approach selectivity, placing the responses to texture stimuli as the centerpiece of the present work according to the reviewers’ helpful suggestions.

[Editors' note: further revisions were suggested prior to acceptance, as described below.]Essential revisions:The range of texture stimuli: As discussed in the first review one might expected that when texture stimuli become too coarse (covering the RGC antagonistic surround), receding motion is preferred. The report states that stimuli begin at 0.8 cycles / degree, the approximate width of the ganglion cell dendritic tree. Are there results from lower spatial frequencies, where one might expect the bias to reverse? If not, please test this range in the model and mention in the Results section.

As requested, we have now tested lower spatial frequency ranges on our computational model and the results are presented in Figure 4D, E. As the reviewers suggested, the ability to distinguish the direction of texture motion falls off at very low spatial frequencies (<0.25 cycles/degree). To put this into context, this means that the highest frequencies (i.e., smallest scales) in an image/environment would need to be larger than the size of a golf ball held at arm’s length in order for the direction of texture motion to become ambiguous from the responses of parasol ganglion cells. Thus, we feel that parasol cells and the underlying circuit mechanisms could contribute to approach detection/optical flow across a broad range of environmental conditions, particularly given the statistical properties of natural scenes (Field, 1987).

The coupled subunit model: More detail is needed here, for example the assumed gain and length constant of electrical coupling. How do these assumed parameters relate to known properties of electrical coupling? Is the computed approach bias robust over a range of parameters? Also, given that the electrical coupling occurs before the bipolar output nonlinearity (subsection “Approach motion selectivity present for moving annuli”), this is formally equivalent to assuming a larger receptive field for the bipolar cell. How certain are we about the size of these receptive fields, and how much were they enlarged by your assumed coupling? Clarify in the Materials and methods section and Results section.

As the reviewers aptly pointed out, electrical coupling will undoubtedly expand the size of bipolar cell receptive fields. However, it is unclear whether this increase in receptive-field diameter can explain the observed bias for approaching annuli. To answer this question, we varied subunit receptive-field diameter in models lacking electrical coupling between subunits.

We used a receptive-field diameter of 32 μm in our coupled subunit model that exhibited approach selectivity to moving annuli (Figure 5C). This value was based on a previous study in which we used direct measurements of excitatory synaptic currents in parasol ganglion cells to determine the subunit size, coupling gain between subunits (gain, 0.1), and space constant for electrical coupling in the diffuse bipolar cell networks (λ, 36.4 μm) (Manookin et al., 2018). These model parameters would expand the subunit receptive field by ~7 μm. Thus, if approach selectivity were a result of subunit receptive-field expansion, we would expect to observe comparable approach selectivity values for the coupled model and a model lacking electrical coupling with subunit diameters of ~39–40 μm. However, this was not the case (Figure 5E, F). The coupled subunit models showed approach selectivity values of ~0.5 when the coupling gain was 0.1 (On model, 0.56; Off model, 0.46), but the same models that lacked coupling and with subunit diameters of 40 μm exhibited much lower approach selectivity (enlarged subunit model: On model, 0.09; Off model, 0.03). In fact, doubling the subunit receptive-field diameters did not reproduce the level of approach selectivity observed in the coupled subunit model or in our direct recordings from parasol ganglion cells (Figure 5E, F). Thus, approach bias does not arise primarily from enlarging subunit receptive fields.

What, then, accounts for the observed approach bias? To answer this question, we created a simplified subunit model which is now presented in Appendix 1. Briefly, this model demonstrates that the pattern of subunit activation (via coupli.ng) and the shape of the input-output nonlinearity generate a bias for approaching motion.

The nonlinear center-surround subunit model: It still is not clear why this model fails to reproduce the approach bias for annuli (Figure 5C). For approaching motion, the center will be activated before the surround. In this case the inhibition from wide-field amacrine cells to bipolar cells will be delayed and not affect release. For receding motion, the surround will be activated prior to center stimulation, allowing inhibition to take hold and dampen glutamate release. As mentioned above, the model with electrical coupling is equivalent to one without coupling but with larger BC receptive fields. Why does that make such a difference (Figure 5D)? Additional interpretation of these results will help the reader understand them.

We agree that this point should be clarified. We have added a more detailed description of why the nonlinear subunit model was not strongly suppressed by receding annuli to the text. The receptive-field profiles for example On and Off parasol cells are shown in Figure 5B.

These profiles were obtained via direct measurements as described in Figure 2. The gray regions indicate the widest extent of the annulus stimulus used in the parasol cell experiments and computational models. Thus, the annulus stimulus typically fell within the receptive-field center region with minimal effect on the surround. This allowed us to utilize our computational models to investigate the potential contributions of different circuit components (e.g., electrical coupling between bipolar cells, output nonlinearities) to processing different types of motion without worrying too much about the effects of the receptive-field surround. This is now described in greater detail in Figure 5 and the accompanying text.

The Materials and methods section on receptive field models is hard to understand (tested by one of our PhD students who was given the assignment to replicate the models). For example, what is the subunit's spatio-temporal receptive field in Equation 10. It would be useful if you took the reader through each of the models from start (photons) to finish (spikes) with one continuous series of equations.

We apologize that our description of the model was unclear. Based on these comments, we have extensively reorganized and rewritten the Materials and methods section dedicated to the subunit models. In addition, we have provided a simplified subunit model and example code for generating the model in MATLAB (see Appendix 1).